# The Unreasonable Effectiveness of Entropy Minimization in LLM Reasoning

**Shivam Agarwal, Zimin Zhang, Lifan Yuan, Jiawei Han, Hao Peng**
University of Illinois Urbana-Champaign
{shivama2, haopeng}@illinois.edu

## Abstract

Entropy minimization (EM) trains the model to concentrate even more probability mass on its most confident outputs. We show that this simple objective alone, without any labeled data, can substantially improve large language models' (LLMs) performance on challenging math, physics, and coding tasks. We explore three approaches: (1) EM-FT minimizes token-level entropy similarly to instruction finetuning, but on unlabeled outputs drawn from the model; (2) EM-RL: reinforcement learning with negative entropy as the only reward to maximize; (3) EM-INF: inference-time logit adjustment to reduce entropy without any training data or parameter updates. On Qwen-7B, EM-RL, without any labeled data, achieves comparable or better performance than strong RL baselines such as GRPO [68] and RLOO [1] that are trained on 60K labeled examples. Furthermore, EM-INF enables Qwen-32B to match or exceed the performance of proprietary models like GPT-4o, Claude 3 Opus, and Gemini 1.5 Pro on the challenging SciCode benchmark [78], while being 3x more efficient than self-consistency and sequential refinement. Our findings reveal that many pretrained LLMs possess previously underappreciated reasoning capabilities that can be effectively elicited through entropy minimization alone, without any labeled data or even any parameter updates. [1]

## 1 Introduction

Entropy minimization (EM) trains the model to concentrate even more probability mass on its most confident outputs without any labeled data or external supervision [23, 60, 25]. It operates on a key assumption and a simple intuition: if a model is reasonably capable, it is more likely to be correct when it is confident [56]. Modern large language models (LLMs) are pretrained on massive data and often undergo a dedicated annealing phase to enhance downstream task performance [59, 3, 92, 77]. It should be, therefore, fairly reasonable to treat them as capable models even before any supervised finetuning (SFT) or reinforcement learning (RL) [90]. This naturally invites the questions: *Can entropy minimization alone improve their performance? And how far can they go without any labeled data?* This work answers both: *yes, and surprisingly far, at least for reasoning tasks.*

We present a systematic exploration of post-pretraining adaptation and inference-time scaling of LLMs, using entropy minimization alone without any labeled data. We propose three novel methods, each aligned with an established post-pretraining stage.

(1) **Unsupervised finetuning by directly minimizing token-level entropy (EM-FT)** mirrors SFT and minimizes a token level loss, on unlabeled outputs sampled from the model conditioning on the input prompts [46]. We find that EM-FT achieves surprisingly strong performance on math and coding tasks, and can even outperform labeled GRPO and RLOO on LeetCode [26] (coding) and Minerva [42] (math).

---

[1]Code:https://github.com/shivamag125/EM_PT

39th Conference on Neural Information Processing Systems (NeurIPS 2025).

(2) **Reinforcement learning with a negative entropy reward (EM-RL)** uses a reward signal based solely on entropy: the negative sum of token-level entropy across a rollout, adjusted by a constant baseline. This is analogous to the REINFORCE algorithm [76, 1] but with entropy as the only supervision without any labeled data. We find that without any labeled data EM-RL can achieve competitive performance to RLOO and GRPO on most math and coding tasks while outperforming it on LeetCode, Minerva and AMC (math) [43].

(3) **Inference-time scaling through entropy minimization (EM-INF)** optimizes the logits during each decoding step to reduce the entropy of the LLM's distribution without any parameter update. We find that EM-INF works best in complex tasks with high uncertainty (e.g. AIME math [43], UGPhysics [88] and SciCode [78]). We observe that Qwen 32B [77] can outperform frontier models like GPT-4o on Scicode [78] and is 3x more efficient than inference scaling through self-consistency and sequential refinement.

While the results are promising, they must be interpreted with care (§5). First, EM is most effective when model confidence correlates with correctness, as in the experiments above. It is less suited for tasks like aligning with human values [35], where confidence alone is not a reliable proxy for quality. Through experiments on individualistic value reasoning [35], we observe that Qwen-2.5 struggles on the task without finetuning on labeled data, and further EM leads to no improvements. Second, the effectiveness of EM hinges on the assumption that the pretrained model is already capable in the tasks of interest. We observe that the improvements from EM-FT and EM-RL on Llama-3.1-8B are less substantial on math reasoning tasks compared to those on Qwen-2.5. This may be due to Llama-3.1-8B is less capable on these reasoning tasks.

The success and limitations of EM highlight the importance of the capabilities of the pretrained models, which is sometimes underappreciated, at least for reasoning tasks. We call for the inclusion of EM as a baseline in the evaluation of future post-pretraining and inference-time scaling algorithms, to better separate the contribution of algorithmic innovations from the intrinsic capabilities of the model itself. Besides, as we show, an approach that performs surprisingly well on certain models and tasks can fail in trivial ways on others. Broad evaluation across diverse pretrained models and tasks can help ensure that the conclusions generalize and are not artifacts of a specific model or dataset.

## 2 Background and Related Work

### 2.1 Related Work

**Entropy-regularized RL** augments the expected reward objective with a regularization term aiming to *increase* entropy [98], in order to promote exploration by adding uncertainty into the actions [64, 33, 27, 18, 41]. Entropy-regularized RL usually learns a policy from scratch in grid world situations (e.g. robot simulations) where adding perturbations can lead to exploring various paths to get to the goal. In contrast, we investigate if *reducing* the uncertainty of the policy can improve a pretrained language models capabilities, with the hypothesis that they already acquire some capabilities for the tasks of interest during pretraining. Instead of learning from scratch, we finetune LLMs to reinforce confident outputs. Therefore, we draw inspiration from entropy-regularization in RL to design reward function (EM-RL) that minimize the entropy of the model encouraging it to commit to a fewer more confident paths, instead of prompting exploration like in previous works.

**Entropy minimization (EM)** has been extensively used as regularization in semi-supervised learning [23, 40, 8, 51], domain adaptation [60, 62, 80], and few-shot learning [25, 39]. [38, 20] apply EM for reinforcement learning with semi-supervised scenarios, where entropy is used to estimate the rewards on unlabeled data. [81, 94, 56] use EM for unsupervised test-time adaptation, where pre-trained classifiers are adapted to new data at test time without using any labels. Despite its well-studied history [53, 52, 48, 15, 61, 71], EM has rarely been explored as a standalone training objective for LLMs. Our key hypothesis is that EM can reinforce LLMs' strong capabilities acquired during pretraining. Instead of applying entropy minimization for test time-domain adaptation or as a regularizer, we study if it can be used for LLM finetuning and inference time scaling.

**Unsupervised and few-shot RL for LLMs** has raised the possibility of self-improvement [85, 5, 54]. Here a model learns via self-training in which it critiques its own generations without external feedback [22, 47, 57, 93] or refines generated text [11, 96, 70]. Building on self-improvement, self-verification [45] has been applied for RL tuning [95, 84, 99], concurrently. [32] uses the most

frequent answer as reward (majority voting) for finetuning, since optimizing the likelihood of most consistent answer can reinforce pretraining priors. [99] applies this idea to unsupervised RL tuning at test time by optimizing on test dataset as a whole. We work on the same intuition of reinforcing model confidence, however, we study post-training where EM-FT and EM-RL are used for finetuning LLMs on large general math and coding data. Further, consistent answers and output extraction is not always possible in complex tasks like scientific coding [2, 10, 82, 4, 73]. On the other hand, EM-FT and EM-RL can be applied on tasks like code generation where output extraction is non-trivial. Unlike [99] which updates the model parameters on each test set individually, inference scaling via EM-INF optimizes the models' entropy without updating parameters and does not need task-specific optimization. EM-INF can be deployed and served like regular text decoding [30, 58, 19] requiring neither assumption on task nor model optimization. [95] proposes to estimate rewards by computing the frequency of extracted answers for a prompt. Here, the model is incentivized to prefer certain answers, which makes the answer distribution more concentrated. While it reduces the entropy, it is not directly enforced and emerges from learning to repeat most frequent solutions. EM-FT and EM-RL target uncertainty in models' token level distribution, whereas the frequency-based approaches target alignment with desired outputs where entropy reduction is not an inherent outcome and task-dependent. [84] investigates RL from few-shot output verified labeled data and observe that direct entropy *maximization* as a standalone objective can be used for finetuning to a small extent. They find that carefully selecting a few output-verifiable problems along with entropy maximization as a regularizer can lead to improvements. We find that only negative entropy rewards (entropy minimization) are sufficient for RL with similar effects without requiring any data selection. Standalone entropy maximization does not converge and eventually collapses [84], but EM-RL and EM-FT via EM converge and can be used to train on large datasets without output verification.

## 2.2 Preliminaries

**Problem setup:** Let $\pi_\theta$ denote an autoregressive LLM policy: $\pi_\theta(\boldsymbol{y}|\boldsymbol{x}) = \prod_i \pi_\theta(y_i|\boldsymbol{x}, \boldsymbol{y}_{<i})$, where $\boldsymbol{x}$ denotes the input prompt and $\boldsymbol{y} = y_1 \ldots y_{|\boldsymbol{y}|}$ the output trajectory. For clarity, we will suppress the conditioning on $\boldsymbol{x}$ whenever it is clear from the context. Let $r(\boldsymbol{y})$ denote a reward function, which provides a score for an output sequence. RL learns a policy that produces output sequence to maximize the expected reward $\mathbb{E}_{y \sim \pi_\theta}[r(\boldsymbol{y})]$.

Entropy minimization minimizes the Shannon entropy $\mathcal{H}(\pi_\theta) = -\mathbb{E}_{\boldsymbol{y} \sim \pi_\theta}[\log \pi_\theta(\boldsymbol{y})]$ [67] of the policy [23]. We consider two methods for empirically estimating the entropy.

**Trajectory-level entropy estimator** [79, 17] is computed based on the trajectory distribution of the output sequences produced by the policy. In practice, it samples full sequences and computes the total log-probability. For an autoregressive policy,

$$\hat{\mathcal{H}}_{\text{traj}}(\pi_\theta) = -\frac{1}{N} \sum_{i=1}^{N} \log \pi_\theta\left(\boldsymbol{y}^i\right) = -\frac{1}{N} \sum_{i=1}^{N} \sum_{t=1}^{|\boldsymbol{y}^i|} \log \pi_\theta(y_t^i|\boldsymbol{y}_{<t}^i) \tag{1}$$

$\{\boldsymbol{y}^i \sim \pi_\theta\}_{i=1}^N$ are $N$ full trajectories sampled from the policy.

**Token (action) level entropy estimator** [27, 18] computes the entropy of the policy based on the token (action) distribution at each step. In practice, it sums per-token entropy across time steps. At generation step $t$, the model produces a probability distribution $\pi_\theta(\cdot|\boldsymbol{y}_{<t})$ over a vocabulary $\mathcal{V}$. The entropy of the policy using token-level estimation is then

$$\hat{\mathcal{H}}_{\text{tok}}(\pi_\theta) = \frac{1}{N} \sum_{i=1}^{N} \sum_{t=1}^{|\boldsymbol{y}^i|} \mathcal{H}\left(\pi_\theta(\cdot \mid \boldsymbol{y}_{<t}^i)\right) \tag{2}$$

where $\mathcal{H}(\pi_\theta(\cdot \mid \boldsymbol{y}_{<t})) = -\sum_{j \in \mathcal{V}} \pi_\theta(j \mid \boldsymbol{y}_{<t}) \log \pi_\theta(j \mid \boldsymbol{y}_{<t})$. While both trajectory-level and token-level estimators are used to compute entropy, they lead to different behaviors when used for training. Minimizing trajectory entropy leads to policies with lower entropy over trajectories whereas minimum token-level entropy leads to policies with low entropy at each generation step. Furthermore, the token-level estimate is commonly used in maximum entropy-regularized RL frameworks, such as soft actor-critic due to its low variance [14, 27, 64, 41].

In the following subsections, we introduce three post-training methods based on entropy minimization as a standalone objective. (1) EM-FT (§3.1): directly minimizes the token-level entropy (Eqn. 2)

similar to supervused finetuning, but on unlabeled outputs drawn from the model; (2) EM-RL (§3.2): reinforcement learning with negative trajectory-level entropy (Eq. 1) or neagtive token level entropy (Eqn. 2) as the only reward to maximize; (3) EM-INF (§4): inference-time logit adjustment to reduce entropy using the token level estimate (Eqn. 2) without any training data or parameter updates.

# 3 Entropy Minimization for Post-training

## 3.1 EM-FT: Direct Entropy Minimization through Token-Level Finetuning

Supervised finetuning methods such as rejection sampling finetuning [16, 91, 46] sample data from the policy and use a reward function to select high-quality data. Good performance on these tasks has been attributed to post-training labels (e.g. output supervision) and pretraining task mixtures [65, 89]. Building on entropy minimization and on-policy finetuning we test if directly minimizing the entropy of the sampled trajectories can enable learning without labels. Given a batch of $N$ sampled trajectories $\{\boldsymbol{y} \sim \pi_\theta\}_{i=1}^N$, we directly minimize the the empirical total token-level entropy $\hat{\mathcal{H}}_{\text{tok}}$ (Eqn. 2). We present this objective and its corresponding gradient in Table 1.

## 3.2 EM-RL: Entropy Minimization with Negative Entropy Rewards in RL

We now minimize the entropy of the policy by using negative entropy as the *only* reward in RL (EM-RL). We design a reward function to assign outcome rewards to trajectory $\boldsymbol{y}$ based on: 1) negative trajectory level entropy (EM-RL-sequence); 2) negative token level entropy (EM-RL-token).

**EM-RL-sequence** minimizes the entropy based on the trajectory level estimator of entropy (Eqn. 1). This estimate favors trajectories that receive high probabilities but does not care about if the reasoning paths are concentrated or not to a few options. Minimizing this entropy estimate is useful in situations where multiple but limited chains of thought are preferred, for example, shorter math tasks where some plausible solutions can be explored but not too many. We use the joint probability of the sequence as a reward $r_{\text{traj}}$ to minimize the trajectory level entropy estimate in expectation, $\mathbb{E}_{\boldsymbol{y} \sim \pi_\theta}[r_{\text{traj}}(\boldsymbol{y})] = -\hat{\mathcal{H}}_{\text{traj}}(\pi_\theta)$. The reward $r_{\text{traj}}$ is,

$$r_{\text{traj}}(\boldsymbol{y}) = \sum_{t=1}^{|\boldsymbol{y}|} \log \pi_\theta(y_t|y_{<t}) = \log \pi_\theta(\boldsymbol{y}) \tag{3}$$

**EM-RL-token** minimizes the entropy based on the token level entropy estimate. This estimate prefers trajectories that are more deterministic and confident at each generation step, focusing the probability mass on fewer possible reasoning paths (chains of thought). For example, problems requiring complex long chains of thought where being more deterministic at every step helps prevent the model from losing track of the reasoning process. Based on the token level entropy estimate (Eqn 2), we define the reward $r_{\text{tok}}$ to minimize the token level entropy estimate in expectation, $\mathbb{E}_{\boldsymbol{y} \sim \pi_\theta}[r_{\text{tok}}(\boldsymbol{y})] = -\hat{\mathcal{H}}_{\text{tok}}(\pi_\theta)$,

$$r_{\text{tok}}(\boldsymbol{y}) = -\sum_{t=1}^{|\boldsymbol{y}|} \mathcal{H}\big(\pi_\theta(\cdot \mid y_{<t})\big) \tag{4}$$

While both the objectives EM-FT, EM-RL aim to minimize the entropy of the policy, they are optimized differently. EM-FT minimizes the entropy by directly differentiating it, whereas EM-RL uses policy gradients [64, 1]. Table 1 summarizes the objectives and the corresponding gradients. In practice, we subtract the RLOO baseline [1] in EM-RL, which does not change the expectation of the gradients but helps with optimization [75, 49, 87]. In addition, we use a KL-regularizer weighted by $\beta$ between the policy $\pi_\theta$ and the initial base model $\pi_{\text{ref}}$ to avoid large deviations from the base model and aid optimization, i.e., $-\mathcal{H}(\pi_\theta) - \beta\text{KL}[\pi_\theta||\pi_{\text{ref}}] = -\mathcal{H}(\pi_\theta) - \beta(-\mathcal{H}(\pi_\theta) - \mathcal{H}(\pi_\theta, \pi_{\text{ref}}))$ [64, 74]. The KL regularization can increase the entropy when the base model is too uncertain (higher entropy), however when $\beta < 1$ (we use 0.001) the dominant gradient is from the reward, in our case entropy minimization.

**Connecting to entropy regularized RL:** Entropy-regularized RL by previous works [27, 18] uses a an entropy maximization term (as a regularizer) along with the reward signal (policy loss) to

Table 1: Comparing various entropy minimization estimates and their corresponding gradients. $N$ denotes the number of trajectories. Min. denotes minimizing an objective and Max. maximizing.

| | | Objectives | Gradients wrt. $\theta$ |
|---|---|---|---|
| Finetuning | **EM-FT** §3.1 | Min. $\hat{\mathcal{H}}_{\text{tok}}$ | $\frac{1}{N}\sum_{i=1}^{N}\sum_{t=1}^{|y^i|}\nabla_\theta \mathcal{H}\left(\pi_\theta(\cdot \mid \boldsymbol{y}_{<t}^i)\right)$ |
| RL | **EM-RL-sequence** §3.2 | Max. $-\hat{\mathcal{H}}_{\text{traj}}$ | $\frac{1}{N}\sum_{i=1}^{N}\left[\sum_{t=1}^{|y^i|}\log \pi_\theta(y_t^i \mid \boldsymbol{y}_{<t}^i)\nabla_\theta \log \pi_\theta(\boldsymbol{y}^i))\right]$ |
| RL | **EM-RL-token** §3.2 | Max. $-\hat{\mathcal{H}}_{\text{tok}}$ | $\frac{1}{N}\sum_{i=1}^{N}\left[-\sum_{t=1}^{|y^i|}\mathcal{H}(\pi_\theta(\cdot \mid \boldsymbol{y}_{<t}^i)\nabla_\theta \log \pi_\theta(\boldsymbol{y}^i))\right]$ |
| Inf-Scaling | **EM-INF** §4 | Min. $\hat{\mathcal{H}}_{\text{tok}}$ | N/A |

encourage stochasticity in action selection and promote exploration. [84] build on this idea and use few labeled examples for the policy loss along with entropy maximization. They find that using entropy maximization *only* can boost performance initially, however, the optimization with only entropy maximization does not converge and eventually collapses the policy. In contrast, EM-RL aims to reinforce pretraining biases by learning a more deterministic policy via entropy minimization without requiring any labeled examples. We find that EM-RL can converge on large datasets and improve model performance on math and code generation.

## 3.3 Training Setup

We use the following setup for all experiments, and provide additional details in Appendix D.

**Evaluation:** For our training experiments, we evaluate on math and coding tasks using [13]- Math-500 [29], AMC [43], AIME 2024 [43], Minerva math (Minerva) [42], Olympiad Bench (**Olymp.**) [28], LeetCode (**LeetC**) [26], and LiveCodeBench-v2 (**LiveC**) [34]. For inference time scaling, we additionally evaluate on **Scicode** [78] and **UGPhysics** [88]. Following [63, 37], we estimate the training FLOPs as $6PD$ and inference FLOPs as $2PD$ where $P$ is the number of parameters in the model and $D$ is the number of tokens (Appendix §D.4)

**Training data, models, and methods:** We construct the training data for math and coding by randomly sampling 35K prompts from Numina math [43] and 25K prompts from Eurus-2 coding split [13], respectively. We use Qwen2.5-Math-7B [89] and Eurus-2-7B-SFT [13] model for training on math and coding, respectively. We use Verl [69] to train. For all RL methods, we use early stopping based on the validation set. We set the number of rollouts to $N = 4$, batch size to $512$, learning rate to $1e^{-6}$. We use a KL regularizer with small coefficient $\beta = 0.001$ which does not effect entropy minimization. We use RLOO and GRPO with output verification only without reward models. We use the same prompts for EM-FT but without the labels. Instead it samples solutions to estimate the entropy. All experiments are done using 4xGH200 Nvidia GPUs.

**Baseline:** We consider an additional rewarding scheme based on the frequency of the final answer–**Self consistency (SC-RL)**. It assigns rewards to each trajectory $\boldsymbol{y}_i$ based on majority voting [85, 32]. An answer $a_i$ is extracted for the trajectory $\boldsymbol{y}_i$ and the frequency of the answer is used estimate the reward, $r(\boldsymbol{y}_i) = \sum_{j=1}^{N}\frac{\mathbb{I}(a_i = a_j)}{N}$, where $\mathbb{I}$ is the indicator function. Self-consistency assigns hard rewards to trajectories where answer extraction is possible (e.g. math), however is inapplicable on tasks like code generation. Such rewards are useful on tasks where several consistent answers can be generated (e.g. MATH-500). However, challenging prompts (e.g. scientific coding) often admit multiple plausible solutions with no answer extraction possible, consequently making majority voting inapplicable. [99] apply this idea for training on the test set, however, we use it on the training set.

## 3.4 Results: Entropy Minimization for Post-Training

Table 2 compares the performance of using entropy minimization as a standalone post-training objective to supervised finetuning and output-verified RL.

**Can unsupervised finetuning by directly minimizing token-level entropy improve LLMs?** We first observe that directly minimizing the entropy (EM-FT) can improve improve performance over the base model by $8\%$ on average on math and coding without using any labels. This observation indicates that pretraining priors can be further capitalized to squeeze more performance and entropy can

Table 2: Performance comparison of unsupervised finetuning (EM-FT) and various rewarding methods in EM-RL with supervised finetuning and RL. *Italics*, **Bold** indicates performance improvement over GRPO and SC-RL (self-consistency RL), respectively. Dash line ("–") denotes that self-consistency is inapplicable. FLOPs are reported as $10^{17}$ (§D.4).

| | **Math** | | | | | | **Coding** | | | |
| --- | --- | --- | --- | --- | --- | --- | --- | --- | --- | --- |
| | Math | AMC | AIME | Minerva | Olymp. | Avg. | LeetC | LiveC | Avg. | FLOPs |
| Qwen2.5-7b | 43.8 | 31.3 | 15.6 | 14.7 | 19.0 | 24.9 | 26.1 | 18.4 | 22.3 | – |
| *Trained using 60K labeled prompts* | | | | | | | | | | |
| w/ SFT N=1 | 48.2 | 30.2 | 10.0 | 17.6 | 22.4 | 25.7 | 18.3 | 18.3 | 18.3 | 1.0 |
| w/ RLOO N=4 | 73.0 | 57.8 | 23.3 | 31.2 | 34.2 | 43.9 | 28.3 | 26.7 | 27.5 | 13.1 |
| w/ GRPO N=4 | 71.8 | 56.6 | 21.1 | 25.0 | 35.9 | 42.1 | 25.0 | 25.8 | 25.4 | 13.1 |
| *Our unsupervised methods trained on 60K unlabeled prompts* | | | | | | | | | | |
| EM-FT N=1 | 67.2 | 51.8 | 14.4 | *33.1* | 34.4 | 40.2 | *28.3* | 17.2 | 22.8 | 1.0 |
| SC-RL N=4 | 73.2 | 51.8 | 15.6 | *26.1* | 36.7 | 40.7 | – | – | – | 13.1 |
| EM-RL-SEQUENCE N=4 | 67.2 | **53.0** | **21.1** | ***30.9*** | 35.6 | **41.6** | *31.1* | 21.7 | *26.4* | 13.1 |
| EM-RL-TOKEN N=4 | 70.8 | ***57.8*** | **18.9** | ***30.9*** | 35.9 | **42.9** | *29.5* | 24.5 | *27.0* | 13.1 |

independently contribute to performance gains. We also note that with only $N = 1$ trajectory sampled from the model, EM-FT can achieve competitive performance to GRPO and RLOO (outperforms on Minerva and LeetCode by $5\%$ on average) which require output verification over multiple sampled ($N = 4$) trajectories. EM-FT better contextualizes the improvements from supervised methods and better uses pretraining capacity, while maintaining a smaller computational footprint. We also observe no improvements on LiveCode likely because LiveCode is regularly updated with new problems that might be different from the pretraining distribution [34]. This observation indicates that entropy minimization can potentially boost performance on unlabeled scenarios if it's not significantly different from the pretraining task mixture [81, 73, 31].

**Can we improve learning from entropy minimization through RL?** In Table 2, we observe that unsupervised RL methods (SC-RL,EM-RL) improves base model performance by $11\%$ on average and maintains competitive performance as compared to output-verified GRPO and RLOO (outperforms on AMC, Minerva and LeetCode by $4.5\%$ on average) under similar compute constraints. This observation suggests that model confidences in the form of token and sequence level entropy estimation can be used as pseudo-rewards to learn from unlabeled scenarios and better use pretraining priors. This observation also ties up with [55, 20, 31] who show that model confidences can be a proxy for reward. Among the various rewarding schemes, we observe that token level and sequence level entropy estimation can be flexibly applied to a variety of tasks without assumptions on the problem structure (e.g. answer extraction) unlike self-consistency which is not applicable for code generation. On average, we observe that EM-RL has better performance via entropy rewards, indicating that fewer deterministic CoTs is more useful on certain reasoning tasks (LiveCode, AMC). We also notice that EM-RL improve model performance on LiveCode by $6\%$ where direct entropy minimization (EM-FT) did not work, indicating that careful reward design to learn from unlabeled experience can better enhance learning and pretraining capabilities. Through EM-FT and EM-RL, we show that EM can elicit pretraining priors necessary for reasoning without any labeled data. While supervised training is expected to scale better, we put forward EM-FT and EM-RL as important baselines to better contextualize improvements from post-training methods.

## 4 EM-INF: Inference-time Logit Optimization

Inference-time scaling methods suggest that enabling LLMs to improve their output generations at test time by scaling compute is critical for improving performance on a complex tasks [86, 72, 66, 50, 6]. Practical LLMs need to adapt to tasks online, i.e., we should be able to adapt to user-queries while the requests are being received. As a result, online LLM adaptation should be unsupervised. Self-consistency (parallel generations) [83] and iterative-refinement (sequential generations) are widely used for reasoning when external verification is not available [47]. Self-consistency samples multiple answers from the model and then selects the solution with the largest frequency. Whereas

Table 3: Performance comparison of EM-INF against test-time scaling methods. **Bold** & *Italics* indicates best and second best performance, respectively. We use a temperature of $t = 0.1$, sample $n = 1$ trajectory for all methods unless specified and report the average of three runs.

| | **Math** | | | | | | **Coding** | **Science** |
|---|---|---|---|---|---|---|---|---|
| | **Math** | **AMC** | **AIME** | **Minerva** | **Olymp.** | **Avg.** | **LeetCode** | **UGPhysics** |
| Qwen2.5-Math-7B-Instruct | 80.7 | 50.6 | 11.9 | 35.7 | **41.1** | 44.0 | 0.6 | 18.5 |
| Greedy Decoding ($t = 0$) | 79.0 | 49.4 | 10.0 | 34.6 | 39.9 | 42.6 | *1.1* | 18.8 |
| Iterative-refinement ($N = 3$) | 73.0 | 39.7 | 7.5 | 32.4 | 33.2 | 37.2 | 0.4 | 17.4 |
| Self-consistency ($N = 4$) | 80.7 | 50.9 | *13.3* | **36.7** | *41.0* | *44.5* | 0.8 | 18.5 |
| ADAPTIVE TEMP | **81.9** | *53.1* | 10.7 | *36.3* | 39.8 | 44.4 | **2.2** | **22.5** |
| EM-INF | *81.3* | **55.2** | **14.8** | 34.8 | 40.6 | **45.4** | 0.6 | *21.4* |
| Qwen2.5-7B-Instruct | 74.3 | 44.6 | 9.1 | 35.2 | 38.4 | 40.3 | 47.2 | 22.7 |
| Greedy Decoding ($t = 0$) | 73.2 | 42.2 | 8.8 | 33.8 | 37.9 | 39.2 | 46.1 | 23.7 |
| Iterative-refinement ($N = 3$) | 72.3 | 40.9 | 8.8 | 34.8 | *38.9* | 39.1 | *50.0* | 21.5 |
| Self-consistency ($N = 4$) | *74.4* | 43.7 | *10.0* | 35.1 | **39.2** | 40.5 | 48.8 | 23.6 |
| ADAPTIVE TEMP | **74.9** | **49.8** | *10.0* | **38.4** | 37.5 | **42.2** | 49.4 | *24.1* |
| EM-INF | 74.0 | *46.7* | **11.7** | 36.5 | 38.7 | *41.5* | **51.7** | **25.4** |
| Llama-3.1-8B-Instruct | 41.9 | 19.2 | 3.3 | *21.1* | 16.5 | 20.4 | 13.1 | 17.1 |
| Greedy Decoding ($t = 0$) | 40.6 | 16.9 | 3.3 | 21.0 | 16.0 | 19.6 | 12.8 | 16.1 |
| Iterative-refinement ($N = 3$) | 41.3 | 18.7 | 2.5 | 20.3 | 16.0 | 19.8 | 9.6 | *18.3* |
| Self-consistency ($N = 4$) | *43.3* | 20.2 | 2.8 | 20.1 | **18.1** | 20.9 | 12.8 | 16.9 |
| ADAPTIVE TEMP | 43.1 | *22.7* | 3.3 | 20.0 | *17.3* | *21.3* | **16.3** | **19.9** |
| EM-INF | **43.9** | **23.5** | 3.3 | **21.3** | 16.2 | **21.6** | *14.1* | 17.6 |

iterative-refinement feeds the previously generated solution into the model as in-context examples inorder to refine the solution. However, online adaptation should not be tailored to certain problem types—self-consistency only works when output extraction is possible and iterative refinement is bottle-necked by context length. To tackle these challenges we propose EM-INF that uses entropy minimization at inference time for logit adjustment. EM-INF treats the model's output logits as if they *were* parameters, and uses gradient descent to update them to minimize the entropy of the distribution they induce; no gradient wrt. or the model parameters or parameter update is needed.

Let $z_t \in \mathbb{R}^{\mathcal{V}}$ be a the logit vector produced by the model at generation step $t$ from its last layer. We freeze the model parameters and optimize the logits $z_t$ to minimize the entropy of the output distribution it induces via gradient descent. We treat the logits as free parameters and *never* backpropagate through the model or update its parameters. To prevent over-optimization, which would effectively lead to greedy decoding, we use a minimum entropy threshold $\delta$ (we empirically find $0.1 < \delta < 0.5$ works the best). Our inference time objective for each generation steps is defined as,

$$\mathcal{L}_{\text{EM-INF}} = \max \Big( - \sum_{j \in \mathcal{V}} \sigma(z_t)_j \log \sigma(z_t)_j, \delta \Big), \quad (5)$$

where $\sigma$ denotes the softmax function. After logit optimization, we use sampling based decoding to select the next token, since this optimization does not change the top logit and greedy decoding will not change the output post-optimization (Prop. 1) [31]. The above optimization operates over $z_t$ and is equivalent to optimizing a one-layer neural network with $|\mathcal{V}|$ parameters with a softmax activation. We empirically find that 5 to 15 gradient steps are enough. The vocabulary size ($\sim 150K$ in Qwen2.5 [77]) is much smaller than number of parameters (7B) in the model, making the optimization overhead negligible compared to a forward pass of the full model. We require only $\mathcal{O}(n)$ ($n$ is sequence length) forward passes from the language model, equivalent to regular decoding. Whereas, other inference time scaling methods like self-consistency and sequential refinement require $\mathcal{O}(Nn)$ forward passes where $N$ is the number of trajectories.

**Baseline:** We use adaptive temperature scaling to reduce the entropy of the models' output distribution. Starting from logits $z_t$ we gradually decrease the temperature $\tau$ in the softmax until the entropy reaches a target value. We search for the largest $\tau > 0$ such that $\mathcal{H}(\sigma(z_t/\tau)) \leq \max(\alpha \mathcal{H}(\sigma(z_t)), \delta)$, where $\alpha \in (0, 1)$ controls the fraction of entropy to reduce. We find $\alpha \in (0.4, 0.6)$ works the best.

**Proposition 1.** *(Informal) Both adaptive temperature scaling and logit optimization reduce the entropy of a model's output distribution. However, logit optimization can change the order of non-top*

*logits, while temperature scaling preserves the order of logits and merely sharpens or flattens the distribution proportionally. Both temperature scaling and logit optimization will keep the token with the top logit and increase its probability, making them less likely to change the model's behavior when it is highly confident (Appendix A).*

**EM-INF for scientific coding:** We apply EM-INF to scientific coding in Table 4. SciCode is a challenging code generation benchmark requiring significant domain knowledge and reasoning capabilities where even frontier models such as GPT4o struggle [78]. We observe that test time entropy minimization (EM-INF, adaptive temperature) can improve performance of the base models consistently on easier subproblems and eventually on the more difficult main problem by 2.5%. We note that logit optimization (EM-INF) is better than adaptive temperature scaling on SciCode by 3%. Owing to the difficulty of SciCode, the models' output distribution has large entropy. Under this scenario, EM-INF can change the relative order of the logits which likely leads to better reasoning chains. Whereas, adaptive temperature scaling will never change the relative ordering of the logits on each generation step – it only makes the distribution peakier (Proposition 1). Next, we observe that scaling through iterative-refinement [47] leads to some improvements over the base model, however, it is bottlenecked by the max context length of the model making it harder to refine long chains of thought. Whereas, EM-INF does not require refinement and maintains the same computational

Table 4: Performance comparison of inference time scaling with ADAPTIVE TEMP and EM-INF with API models and inference time scaling methods on Scicode. Sub and main indicate subproblem and main problem mode in Scicode, whereas w/BG refer to with background knowledge in the prompts. **Bold**,*Italics* indicate the best and second best performance over all methods.

|  | Sub w/ BG | Main w/ BG |
| --- | --- | --- |
| GPT-4o | 35.4 | 9.2 |
| GPT-4-Turbo-2024-04-09 | 33.7 | 9.2 |
| Claude3.5-Sonnet | 35.4 | 12.3 |
| Claude3-Opus | 26.7 | 4.7 |
| Claude3-Sonnet | 25.7 | 4.7 |
| Gemini 1.5 Pro | 30.6 | 7.7 |
| Qwen2.5-7b-Instruct | 11.5 | 0.0 |
| Greedy decoding ($t=0$) | 13.4 | **3.0** |
| Iterative-refinement ($n=2$) | 12.1 | 0.0 |
| ADAPTIVE TEMP | *13.2* | 0.0 |
| **EM-INF** | **16.7** | *1.5* |
| Qwen2.5-32b-Instruct | 25.4 | 4.6 |
| Greedy decoding ($t=0$) | 25.4 | 4.6 |
| Iterative-refinement ($n=2$) | 24.7 | 7.6 |
| ADAPTIVE TEMP | **28.8** | *7.6* |
| **EM-INF** | *27.1* | **10.7** |

complexity as the base model when used with regular decoding. We note that EM-INF with Qwen2.5-32B-Instruct can outperform several frontier models on the main problem without requiring refinement or editing model parameters by 3% and over the base model by 6%. Building on these observations, we put forward EM-INF as a practical, compute efficient inference time scaling method that does not require any supervision and makes no assumption on the nature of the problem.

**EM-INF for math and coding:** Table 3 compares the performance of EM-INF for test time scaling. We observe that minimizing the entropy of the model at test time without updating the parameters (EM-INF, adaptive temperature) improves the performance of the base model for almost all tasks and model classes by 3% on average, indicating that EM is an useful method for test time scaling. We also note that EM can be applied to all tasks (math, coding and ugphysics) without assumptions such as output extraction in self-consistency or output scoring. Next, we note that EM competitively performs against self-consistency while outperforming it on tasks requiring significant reasoning capabilities, e.g. UGPysics by 3.6%. Unlike self-consistency which requires sampling multiple trajectories ($N > 1$), EM-INF requires only one trajectory leading to lower computa-

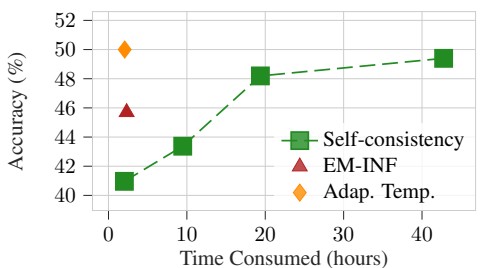

Figure 1: Computational efficiency of EM-INF compared to self-consistency with $N = 4, 8, 16$ on AMC.

tional requirements. These observations collectively indicate that pretraining performance can be improved via EM, and making the model more deterministic at inference time via unsupervised EM in-order to prefer fewer chain of thought options can lead to better performance.

Table 5: The limitations of EM on tasks and models where model confidences do not help. **Left** shows performance of Llama-3.1-8B on math tasks trained with the setup in Sec. 3.3. **Right** compares the performance of Qwen-2.5-7b instruct on individualistic value alignment.

| | Math | AMC | AIME | Minerva | Olymp. | | IndVal |
|---|---|---|---|---|---|---|---|
| Llama-3.1-8B-Instruct | 40.6 | 18.1 | 1.0 | 22.4 | 15.7 | Qwen2.5-7B-Instruct | 65.9 |
| W/ RLOO | 52.0 | 25.3 | 3.3 | 26.8 | 20.7 | EM-FT (N=1) | 66.0 |
| EM-FT (N=1) | 42.2 | 21.7 | 3.3 | 18.8 | 16.4 | EM-RL-token (N=4) | 66.0 |
| EM-RL-token (N=4) | 41.0 | 18.1 | 5.6 | 19.1 | 15.6 | EM-RL-sequence (N=4) | 66.3 |
| EM-RL-sequence (N=4) | 41.6 | 18.1 | 7.8 | 18.4 | 16.6 | | |

**Computational efficiency:** Figure 1 compares the compute requirements of EM-INF with inference scaling baselines. We observe that EM-INF can achieve better performance while using a third of time. This observation suggests that EM-INF is a practical inference scaling method in compute bound scenarios, for online adaption without updating model parameters [81, 9]. EM-INF can also be combined with self-consistency and sequential refinement to scale up (Appendix §C.4).

## 5   Limitations: Dependence on Base Model Capability and Confidence

To contextualize the applicability of EM, we probe its performance by 1) training on tasks with different bias as compared to pretraining; and 2) finetuning models lacking target reasoning behaviors.

Table 5 (right) evaluates EM on individualistic value reasoning (**IndVal**) [35]. Individualistic value reasoning is a difficult alignment task that evaluates LLMs' ability to reason about an individual's value preferences in new situations. Given known statements about a certain individual, the task is to predict if the user would or would not associate with a new situation (measured using accuracy). We observe that Qwen-2.5-7B struggles with the task, likely because LLMs' are biased to coarse values acquired from pretraining which may not generalize to a large pluralistic audience [97, 35]. As expected, we note that EM is not effective in this task, likely because model confidences are incorrectly biased to training priors that do not benefit the target task, as also observed in [99, 31].

Furthermore, Table 5 (left) compares entropy minimization (EM-FT, EM-RL) with supervised RL (RLOO) using the same training data, setup and evaluation as Sec.3.3 with Llama-3.1-8b-Instruct as the base model [24]. We observe that while supervised RL can improve model performance, improvements from entropy minimization is less effective as compared to Qwen-2.5 and can sometimes hurt performance on math reasoning. We tie this observation to [21] who show that Llama models may lack reasoning behaviors initially as compared to Qwen models, as also observed in [84]. The lack of certain reasoning behaviors for math in the initial policy possibly leads to unreliable model confidences and eventually inaccurate entropy rewards [12]. This observation suggests that success of entropy minimization hinges on the assumption that the pre-trained model's initial reasoning behaviors are critical for the capacity of improvement. Overall, the success of EM-FT and EM-RL must be interpreted with care, as it can be less effective depending on the task and base model choice.

## 6   Conclusion

We show that entropy minimization through unsupervised finetuning (EM-FT), RL with standalone negative entropy as reward (EM-RL) and inference time scaling through logit optimization (EM-INF) can improve LLMs on complex math, physics, and coding tasks without using any labeled data. We observe that through EM-FT and EM-RL, Qwen-2.5-7B can achieve comparable or better performance than RL methods like GRPO and RLOO. Through test time logit optimization, EM-INF enables Qwen-2.5-32B to outperform proprietary models like GPT4o on complex scientific coding without training or updating model parameters. Our findings suggest that many pretrained LLMs already possess strong reasoning ability which can be enhanced using EM alone. While successful, EM is less beneficial on tasks and models with different inductive biases as pretraining. We call for the inclusion of EM-FT, EM-RL and EM-INF as a baseline in the evaluation of future post-pretraining and inference-time scaling algorithms, to better attribute the source of improvements in them.

## Acknowledgments and Disclosure of Funding

This research used the DeltaAI advanced computing and data resource, which is supported by the National Science Foundation (award OAC 2320345) and the State of Illinois. DeltaAI is a joint effort of the University of Illinois Urbana-Champaign and its National Center for Supercomputing Applications.

This work was also in part supported by research awards from Apple and the Allen Institute for AI.

Research was also supported in part by US DARPA INCAS Program No. HR0011-21-C0165 and BRIES Program No. HR0011-24-3-0325, National Science Foundation IIS-19-56151, the Molecule Maker Lab Institute: An AI Research Institutes program supported by NSF under Award No. 2019897, and the Institute for Geospatial Understanding through an Integrative Discovery Environment (I-GUIDE) by NSF under Award No. 2118329. Any opinions, findings, and conclusions or recommendations expressed herein are those of the authors and do not necessarily represent the views, either expressed or implied, of DARPA or the U.S. Government.

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

# A   Proof for proposition 1

*(Informal) Both adaptive temperature scaling and logit optimization reduce the entropy of a model's output distribution. However, logit optimization can change the order of non-top logits, while temperature scaling preserves the order of logits and merely sharpens or flattens the distribution proportionally. Both temperature scaling and logit optimization will keep the token with the top logit and increase its probability, making them less likely to change the model's behavior when it is highly confident.*

**Part 1:** Let $p(\boldsymbol{z}) = \text{softmax}(\boldsymbol{z})$, and $\mathcal{H}(p)$ denotes the entropy of $p$; $\mathcal{H} = -\sum_{i=1}^{|\mathcal{V}|} p_i \log p_i$. $\exists \boldsymbol{z} \in \mathbb{R}^{|\mathcal{V}|}, \eta > 0$ and $a, b \in [1, \ldots, |V|]$, such that $\boldsymbol{z}_a < \boldsymbol{z}_b$ and $\boldsymbol{y}_a > \boldsymbol{y}_b$ where $\boldsymbol{y}_i = z_i + \eta p_i (\log p_i + \mathcal{H})$, $i = 1, \ldots, |\mathcal{V}|$.

*Proof:* Choose $\boldsymbol{z}$ with two non-top indices satisfying $p_a < p_b < e^{-(\mathcal{H}+1)}$, and therefore $\boldsymbol{z}_a < \boldsymbol{z}_b$. Consider $\boldsymbol{y}_a - \boldsymbol{y}_b = \boldsymbol{z}_a - \boldsymbol{z}_b + \eta(g_a - g_b)$ where $g_i = p_i(\log p_i + \mathcal{H})$. $\boldsymbol{g}$ is the negative gradient of entropy $\mathcal{H}$ w.r.t. $\boldsymbol{z}$. We show that $g_a > g_b$.

By Mean-Value Theorem $\exists c \in (p_a, p_b)$ such that $g_a - g_b = (p_a - p_b)(\log c + \mathcal{H} + 1)$. We then have

$$c < p_b < e^{-(\mathcal{H}+1)} \implies \log c + \mathcal{H} + 1 < 0 \implies g_a - g_b > 0 \tag{6}$$

Thus, we can set an arbitrary large $\eta$ such that,

$$z_a - z_b + \eta(g_a - g_b) > 0 \tag{7}$$

**Part 2:** $b$ cannot be $\arg\max_i z_i$.

*Proof:* Let $j = \arg\max_i \boldsymbol{z}_i$, i.e. the index of the maximum logit. Since $p_j = \max_i p_i$ we can bound it using the inequality:

$$\mathcal{H}(p) = -\sum_i p_i \log p_i \geq -\log p_j \quad \Rightarrow \quad p_j \geq e^{-\mathcal{H}(p)} \geq e^{-(\mathcal{H}(p)+1)} \tag{8}$$

We now analyze how $g_i$ behaves as a function of $p_i$. Let $g(p_i) = p_i(\log p_i + \mathcal{H})$ and $g'(p_i) = \log p_i + \mathcal{H} + 1$. Thus, $g(p_i)$ is monotonically increasing whenever $g'(p_i) > 0$, i.e. when,

$$p_i > e^{-(\mathcal{H}+1)} \tag{9}$$

So no lower-probability index $i$ can have $g_i > g_j$, i.e., the entropy gradient cannot reduce $\boldsymbol{y}_j$ relative to others. Therefore, $b \neq j$, leading to the following behavior

$$0 < g_a \leq g_b \quad p_a \in (e^{-\mathcal{H}}, p_b] \tag{10}$$

$$g_a < 0 < g_b \quad p_a \in (0, e^{-\mathcal{H}}) \tag{11}$$

# B   Adaptive Temperature Scaling

The algorithm of adaptive temperature scaling for EM-INF is presented in Algorithm 1.

# C   Additional Results

## C.1   EM-RL with DeepSeek-Math-7B-Instruct

To further understand initial model choice, we finetune another base model- DeepSeek-Math-7B-Instruct on the same data and training settings as Qwen-2.5-7B-Math (Section 3.3). Deepseek-7b-Math-Instruct has been finetuned on math-related tokens for 500B tokens on top of Deepseek-coder-7b [68]. We use the instruction tuned version which has been further finetuned using chat data. We fine tune Deepseek-7b-Math-Instruct using entropy minimization (EM-RL) and evaluate it in Table 6. We observe that entropy minimization improves performance over the Deepseek model by 2.5% on average on math tasks.

**Algorithm 1** Inference-time Adaptive Temperature

**Input:** Logits $z_t$; Initial maximum temperature $\tau_{max\_init}$; Initial minimum temperature $\tau_{min\_init} > 0$; Maximum iterations $M$; Tolerance $\epsilon_{tol} > 0$; Target entropy reduction ratio $\alpha \in (0, 1)$; Target entropy threshold $\delta > 0$.
**Output:** Scaled logits $z'_t$.

1: **function** COMPUTEADAPTIVETEMPERATURE($z_t, \tau_{max\_init}, \tau_{min\_init}, M, \epsilon_{tol}, \alpha, \delta$)
2: $\quad$ $P_{initial} \leftarrow \text{Softmax}(z_t)$
3: $\quad$ $H_{initial} \leftarrow \text{Entropy}(P_{initial})$
4: $\quad$ $H_{target} \leftarrow \max(\delta, \alpha \cdot H_{initial})$
5: $\quad$ $\tau_{final} \leftarrow 1.0$
6: $\quad$ **if** $H_{initial} > \delta$ **then**
7: $\quad\quad$ $\tau_{low} \leftarrow \tau_{min\_init}$
8: $\quad\quad$ $\tau_{high} \leftarrow \tau_{max\_init}$
9: $\quad\quad$ **for** $iteration \leftarrow 1$ to $M$ **do**
10: $\quad\quad\quad$ $\tau_{mid} \leftarrow (\tau_{high} + \tau_{low})/2$
11: $\quad\quad\quad$ $P_{current} \leftarrow \text{Softmax}(z_t/\tau_{mid})$
12: $\quad\quad\quad$ $H_{current} \leftarrow \text{Entropy}(P_{current})$
13: $\quad\quad\quad$ **if** $|H_{current} - H_{target}| < \epsilon_{tol}$ **then**
14: $\quad\quad\quad\quad$ $\tau_{final} \leftarrow \tau_{mid}$
15: $\quad\quad\quad\quad$ **break**
16: $\quad\quad\quad$ **else if** $H_{current} < H_{target}$ **then**
17: $\quad\quad\quad\quad$ $\tau_{low} \leftarrow \tau_{mid}$
18: $\quad\quad\quad$ **else**
19: $\quad\quad\quad\quad$ $\tau_{high} \leftarrow \tau_{mid}$
20: $\quad\quad\quad$ **end if**
21: $\quad\quad\quad$ $\tau_{final} \leftarrow \tau_{mid}$
22: $\quad\quad$ **end for**
23: $\quad$ **end if**
24: $\quad$ $z'_t \leftarrow z_t/\tau_{final}$
25: $\quad$ **return** $z'_t$
26: **end function**

Table 6: Performance comparison of various entropy based rewarding methods in EM-RL against RL with output verified rewards using Deepseek-Math-7b-Instruct as the base model. We use the same setup as Qwen-2.5-Math-7B (Section 3.3). **Bold** indicates performance improvement over RLOO.

| | Math | | | | |
|---|---|---|---|---|---|
| | Math | AMC | Minerva | Olymp. | Avg. |
| Deepseek-math-7b-instruct | 42.0 | 16.9 | 19.1 | 12.6 | 22.7 |
| *Trained using 60K labeled prompts* | | | | | |
| w/ RLOO N=4 | 42.4 | 24.1 | 20.6 | 13.2 | 25.1 |
| *Our unsupervised methods trained on 60K unlabeled prompts* | | | | | |
| EM-RL-SEQUENCE N=4 | **42.8** | 21.7 | **21.7** | **14.4** | **25.2** |
| EM-RL-TOKEN N=4 | **43.4** | 18.1 | **22.8** | **13.6** | 24.5 |

## C.2 Sensitivity to KL regularization coefficient $\beta$

We provide results with different $\beta$ for EM-RL token below in Table 7. We observe that a small $\beta$ value is better for entropy minimization because it weighs EM more in the overall loss function.

Theoretically, the KL regularization can increase the entropy when the base model is too uncertain, i.e., it deviates from the pre-trained checkpoint causing it to visit high entropy states. However when $\beta < 1$ (we use $0.001$) the dominant gradient in the policy gradient is from entropy minimization. Since we set $\beta$ to a small value the overall objective minimizes the entropy.

Table 7: Performance variation of EM-RL-token using different KL regularization coefficient, $\beta$. We use Qwen-2.5-Math-7b as the base model with the same setup in Sec.3.3.

| | Math | | | | | |
|---|---|---|---|---|---|---|
| **Model** | **Math** | **AMC** | **AIME** | **Minerva** | **Olymp.** | **Avg.** |
| Qwen-2.5-Math-7b | 43.8 | 31.3 | 15.5 | 14.7 | 19.0 | 24.9 |
| EM-RL-TOKEN $\beta = 1.5$ | 45.0 | 39.8 | 13.3 | 9.6 | 21.2 | 25.8 |
| EM-RL-TOKEN $\beta = 0.5$ | 50.8 | 39.8 | 16.7 | 10.7 | 23.0 | 28.2 |
| EM-RL-TOKEN $\beta = 0.001$ | 70.8 | 57.8 | 18.9 | 30.9 | 19.0 | 39.5 |
| EM-RL-TOKEN $\beta = 0$ | 70.9 | 57.5 | 18.2 | 29.6 | 19.0 | 39.0 |

Table 8: Sensitivity to number of rollouts $N$ of EM-RL-SEQUENCE. We use Qwen-2.5-Math-7b and use the training setup defined in Sec.3.3.

| | Math | | | | | |
|---|---|---|---|---|---|---|
| **Model** | **Math** | **AMC** | **AIME** | **Minerva** | **Olymp.** | **Avg.** |
| EM-RL-SEQUENCE N=4 | 67.2 | 53.0 | 21.1 | 30.9 | 35.6 | 41.6 |
| EM-RL-SEQUENCE N=8 | 71.4 | 54.2 | 13.3 | 31.6 | 35.0 | 41.1 |
| EM-RL-SEQUENCE N=16 | 70.6 | 51.9 | 15.5 | 23.9 | 35.1 | 39.4 |

## C.3    Sensitivity to number of rollouts $N$

We measure the impact of the number of rollouts $N$ on EM-RL-SEQUENCE in Table 8. We observe that EM-RL-SEQUENCE benefits with larger on MATH-500, suggesting that larger number of rollouts might reduce the variance. However, we note that a larger $N$ does not help AMC, Minerva and Olympiad bench, and reduces performance on AIME. Overall, we observe consistent performance of EM-RL-SEQUENCE on sweeping $N$ from 4 to 16. We set $N = 4$ building on existing RL fine tuning works to make sure that the training overhead is consistent [13].

## C.4    Combining EM-INF with Self-consistency and Iterative Self-refinement

We combine EM-INF with self-consistency and iterative self-refinement in Figure 2a,2b. Since, EM-INF only optimizes the logits, it can be plugged into both methods easily. We observe that EM-INF can get the same performance as self-consistency (N=4) while using 3 time less FLOPs on AMC with Qwen 2.5 7B. As expected, EM-INF reduces the diversity of the generation, which causes it's benefits to diminish over iterations. Self-consistency benefits the most when several diverse generations can be produces which lead to the same answer, however, EM-INF reduces diversity through entropy minimization [44, 7] leading to smaller benefits with larger compute. EM-INF is most suitable for inference time scaling on high uncertainty tasks under compute bound scenarios.

## C.5    Qualitative analysis of EM-INF on SciCode

We conduct a qualitative analysis of EM-INF on SciCode through a case study where Qwen2.5-7B-Instruct with EM-INF successfully generates the correct code, while the base model fails. The problem and corresponding model generations are illustrated in Figure 3. The prompt requires generating a symmetric matrix with increasing diagonal values, where each element is modified by the product of a user-defined noise factor and a normally distributed random number, followed by symmetrization via averaging with the matrix's transpose.

The base Qwen2.5-7B-Instruct model produces a partially correct implementation of the `init_matrix` function. It correctly initializes the increasing diagonal and applies the symmetrization formula (`A = (A + A.T) / 2`). However, it fails to apply the noise factor to all elements of the matrix. Specifically, the noise modification is applied only to the off-diagonal elements, leaving the diagonal entries (e.g., `i+1`) unaltered by the stochastic process prior to symmetrization. This error occurs because scientific coding is a difficult task where both coding and domain knowledge is required, therefore making the model's output distribution uncertain.

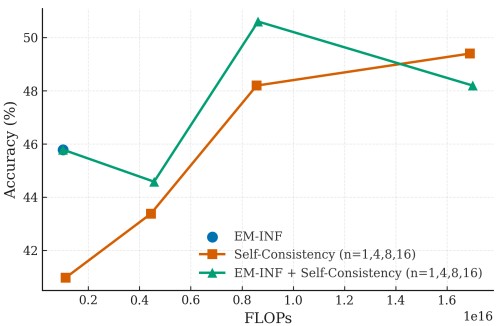

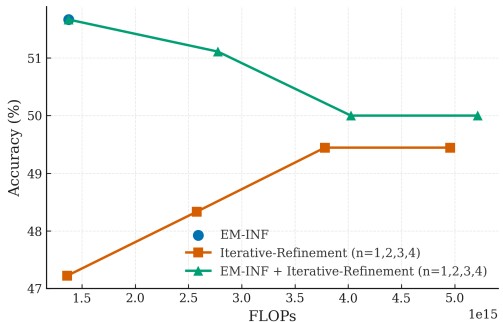

(a) Accuracy vs. FLOPs for combining EM-INF and self-consistency at inference time on AMC.

(b) Accuracy vs. FLOPs for combining EM-INF and iterative refinement at inference time on LeetCode.

Figure 2: Performance of EM-INF combined with various inference-time scaling methods using Qwen2.5-7B-Instruct. EM-INF can effectively improve performance when combining with other inference-time scaling methods while requiring negligible additional computation.

By contrast, the generation using EM-INF produces the correct implementation that adheres to all constraints specified in the prompt. *EM-INF* reduces the entropy of the output distribution, which leads to more deterministic generations. The generated code by EM-INF, as expected, reduces the diversity in generation, leading to more concise implementations and omitting additional comments. These results suggest that EM-INF is particularly beneficial for scientific coding tasks like SciCode, where correctness and determinism are critical.

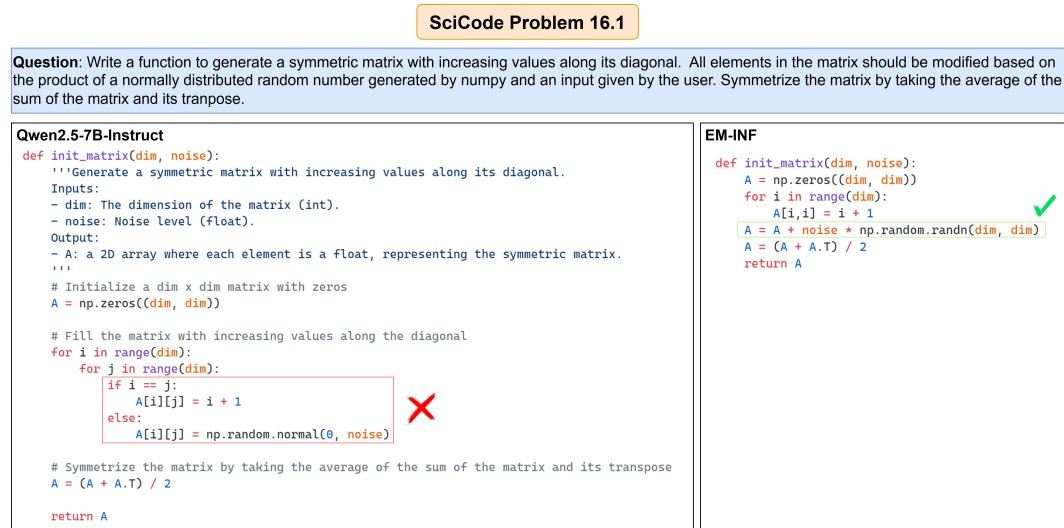

Figure 3: Qualitative analysis of EM-INF on SciCode. Qwen2.5-7B-Instruct's generated code (**Left**) fails to add random noise to all elements in the matrix, whereas the code generated with EM-INF (**Right**) performs the correct operation. We observe that EM-INF (1) reduces the diversity of generated tokens and is more deterministic as expected; (2) generates correct code; (3) produces more concise outputs; and (4) is more helpful in scientific coding tasks like SciCode where LLMs are highly uncertain.

# D  Experimental Setup

## D.1  Evaluation Benchmarks

### D.1.1  Math Reasoning

We adopted the data and evaluation pipeline from PRIME-RL codebase [13] for all math reasoning evaluations. The final answer of LM is the string inside the last $\backslash boxed\{\}$. All math benchmarks is evaluated on Pass@1 accuracy. Here, we will introduce each math benchmarks in detail.

**MATH 500**  MATH 500 contains 500 prompts sampled from the 5,000 test problems in the original MATH dataset [29]. The test set of MATH comprises challenging competition-style math problems drawn from sources such as the AMC and AIME exams. These problems span seven subjects—Prealgebra, Algebra, Number Theory, Counting and Probability, Geometry, Intermediate Algebra, and Precalculus—and are labeled with difficulty levels from 1 (easiest) to 5 (hardest). Each test problem includes a final boxed answer and a full step-by-step solution in LaTeX, enabling exact match evaluation and providing interpretable reasoning paths for model analysis.

**AMC**  We use the validation set from AI-MO's AMC dataset [43], which contains 83 problems extracted from AMC 12 (2022 and 2023). The AMC dataset comprises math competition problems from the American Mathematics Competitions (AMC), covering a broad range of topics including arithmetic, geometry, combinatorics, and algebra. It is designed to evaluate a language model's capacity for mathematical reasoning, abstraction, symbolic manipulation, and logical deduction.

**AIME**  We use the validation set from AI-MO's AIME dataset [43], which includes 90 problems drawn from AIME 22, AIME 23, and AIME 24. The AIME (American Invitational Mathematics Examination) dataset features high-difficulty problems intended for top-performing students from the AMC competitions. These problems typically require advanced mathematical reasoning across topics such as algebraic manipulation, number theory, combinatorics, and geometry. Each problem involves multiple intermediate steps and provides minimal guidance, making the dataset well-suited for evaluating a language model's ability to perform multi-step symbolic reasoning and maintain logical coherence throughout extended problem-solving chains.

**Minerva Math**  Minerva Math refers to the OCWCourses evaluation dataset introduced in the Minerva paper [42]. This dataset comprises 272 undergraduate-level STEM problems sourced from MIT OpenCourseWare (OCW), covering topics such as solid-state chemistry, information theory and entropy, differential equations, and special relativity. Each problem is self-contained and accompanied by a clearly defined final answer—either numeric (191 problems) or symbolic (81 problems)—enabling automatic evaluation. The dataset is designed to assess a language model's ability to perform multi-step scientific reasoning, including the application of domain-specific knowledge, symbolic manipulation, and precise quantitative computation.

**OlympiadBench**  OlympiadBench [28] is a large-scale bilingual multimodal benchmark comprising 8,476 Olympiad-level mathematics and physics problems sourced from international competitions such as the IMO and IPhO, as well as elite Chinese contests and Gaokao mock exams. The dataset evaluates LLMs on advanced mathematical and physical reasoning tasks, including multi-step logical deduction, symbolic manipulation, equation solving, theorem proving, and visual-spatial understanding from diagrams. Its high difficulty, multimodal diversity, and expert-level content make it a rigorous benchmark for assessing scientific reasoning capabilities. In this work, we use the open-ended, English, text-only subset of math competition problems as our evaluation set, which contains a total of 674 problems.

### D.1.2  Coding

**LeetCode**  LeetCode benchmark is introduced in the technical report of DeepSeek-Coder [26]. This benchmark consists of 180 recent competition-level programming problems from LeetCode, collected between July 2023 and January 2024. Each problem is paired with 100 test cases to ensure comprehensive coverage, and prompts are constructed in a standardized format to simulate realistic coding challenges. This dataset evaluates a language model's ability to perform code

generation in competitive programming settings, testing a wide range of capabilities such as problem comprehension, algorithm design, logic reasoning, and implementation precision. The inclusion of varying difficulty levels (Easy, Medium, Hard) allows for a thorough assessment of a model's generalization and problem-solving skills under pressure. During evaluation, the python code will be extracted from model's response and run the test cases. Performance is reported using the Pass@1 metric, indicating the proportion of problems for which the model's first generated solution passes all test cases.

**LiveCodeBench** LiveCodeBench [34] is a large-scale, contamination-free benchmark designed to holistically evaluate the code-related capabilities of large language models. It comprises 612 recent coding problems collected from competitive programming platforms such as LeetCode, AtCoder, and Codeforces between May 2023 and August 2024. Unlike traditional benchmarks that focus solely on code generation, LiveCodeBench assesses four key skills: (1) code generation from natural language, (2) self-repair using error feedback, (3) code execution via output prediction, and (4) test output prediction without implementation. This comprehensive setup evaluates not only functional correctness but also a model's ability to debug, reason about program behavior, and construct tests. Its time-segmented evaluation protocol further supports rigorous detection of training data contamination. In this work, we adopt the V2 release, which includes 511 problems released between May 2023 and May 2024. We evaluate methods on the code generation ability. The model is given a natural language problem description and tasked with generating a correct program. Correctness is evaluated by passing all test cases. The Pass@1 metric is used, measuring the fraction of problems where the first generated solution passes all tests.

### D.1.3 Scientific Reasoning

**UGPhysics** UGPhysics [88] is a large-scale bilingual benchmark designed to evaluate undergraduate-level physics reasoning in large language models. It comprises 5,520 problems (11,040 including both English and Chinese versions) spanning 13 physics subjects and 59 topics, featuring diverse answer formats such as numerical values, symbolic expressions, equations, and intervals. Each problem is annotated with one of four reasoning skill types: Knowledge Recall, Laws Application, Math Derivation, and Practical Application. Unlike prior benchmarks that primarily target multiple-choice or high-school-level content, UGPhysics emphasizes deeper problem-solving and derivation skills, requiring LLMs to perform symbolic reasoning, apply physical laws, and handle real-world quantities. This dataset is designed to assess LLMs' ability to carry out multi-step physics reasoning beyond simple mathematical manipulation. In this work, we evaluate on the Semiconductor Physics subset, which consists of 372 problems. During evaluation, models answer physics problems using prompts tailored to different answer types for improved extraction and judgment. A rule-based correctness assessment then extracts answers from the model's output and reports accuracy.

**SciCode** SciCode [78] is a challenging, scientist-curated benchmark designed to evaluate large language models (LLMs) on real-world scientific coding tasks across 16 subfields in mathematics, physics, chemistry, biology, and materials science. It comprises 80 complex main problems, each decomposed into multiple subproblems (338 in total), requiring models to integrate domain-specific scientific knowledge, reasoning, and Python programming. Each problem includes gold-standard solutions and test cases, with optional expert-authored background context. SciCode assesses LLMs' abilities in multi-step problem decomposition, code synthesis from scientific instructions, and integration of partial solutions. By emphasizing scientific reasoning and simulating real-world workflows, it provides a rigorous benchmark for evaluating LLMs as scientific assistants. In our paper, we evaluate methods on the test set, which contains 65 main problems and 288 subproblems in total. Models generate Python functions for each subproblem and integrate them into a complete solution. Evaluation is fully automatic, using gold test cases for both subproblems and main problems. The evaluation is conducted with and without scientific background information, as well as using gold or generated prior solutions, to measure realistic performance. Success requires all subproblems and the main solution to be correct. Final accuracy is reported as the percentage of problems achieving success.

### D.1.4 Individualistic Value Reasoning

**IndVal**   The INDVAL dataset [36] is derived from the World Values Survey (WVS) and consists of natural language statements expressing individual human values and preferences from over 93,000 participants worldwide. Each participant responds to 253 questions grouped into 13 distinct categories, including Social Values, Economic Values, and Science & Technology, among others. The dataset standardizes these diverse survey responses into over 900 value-expressing statements per individual, enabling evaluation of language models' ability to reason about individualistic human values in a bottom-up, granular manner. This dataset is particularly valuable as it challenges models to understand and predict nuanced personal value judgments beyond demographic stereotypes. Consequently, it reveals limitations in current language models' capacity for individual value reasoning and promotes more equitable, personalized AI alignment that respects individual diversity. In our work, we randomly sampled 200 individuals excluded from training, uniformly distributed across all continents. For each individual, we randomly selected 4 statements per question category to form an individualistic value profile consisting of 52 known statements. Then, for each question category, we selected 2 statements not included in the known profile and asked the language model to select the most appropriate statement from the given candidates based on the individual profile. This process resulted in an evaluation dataset containing 5,400 prompts. During evaluation, the selected statement is extracted from the model generation and compared with the ground truth. Final performance is reported as accuracy.

### D.2   Training Data

**Math Reasoning**   We randomly sampled 35K prompts from Numina math training set [43] as our math training prompts. The NuminaMath dataset contains 860,000 problems ranging from high-school to advanced competition levels, including Olympiad, AMC, AIME, and Chinese K-12 math problems.

**Coding**   We randomly sampled 25K prompts from Eurus-2-RL training dataset's coding split [13] as our coding training prompts. The Eurus-2-RL training dataset is a high-quality reinforcement learning dataset consisting of about 457,000 math problems and 27,000 coding problems with outcome verifiers such as LaTeX answers for math and test cases for coding. The math problems are sourced from the NuminaMath-CoT dataset, which covers a wide range from Chinese high school math to International Mathematical Olympiad-level questions. The dataset is carefully cleaned, filtered, and reformatted into a direct question-answer format to ensure high quality and solvability. It is especially useful for training because it provides diverse, verified, and challenging problems covering different level of difficulties and tasks.

**Individualistic Value Reasoning**   The training dataset is derived from the IndVal dataset [36]. We sampled 900 individuals uniformly across seven continents. For each individual, we randomly selected 4 statements per question category to form an individualistic value profile comprising 52 known statements. Then, we sampled 6 questions per question category as queries, resulting in a total of 70,200 training prompts.

### D.3   Models

**Qwen2.5 Family**   For the entropy minimization for post-training experiments, we use two models derived from the Qwen2.5 family: Qwen2.5-Math-7B [89] and Eurus-2-7B-SFT [13]. Qwen2.5-Math-7B is selected as the base model for math reasoning tasks. We apply various post-training methods using the math training dataset and evaluate their performance on math reasoning benchmarks. This model serves as an excellent starting point for our experiments due to its strong capability in tackling complex mathematical reasoning problems. Eurus-2-7B-SFT is a SFT checkpoint obtained by further fine-tuning Qwen2.5-Math-7B on the Eurus-2-SFT-Data from PRIME-RL [13], which is an action-centric chain-of-thought reasoning dataset. Since Eurus-2-7B-SFT further enhances the reasoning abilities of Qwen2.5-Math-7B, we select it for coding experiments, applying different post-training methods with the coding training dataset and evaluating on coding benchmarks.

For entropy minimization during inference time experiments, we select Qwen2.5-Math-7B-Instruct and Qwen2.5-7B-Instruct as the base models for math reasoning, coding, and scientific reasoning tasks because of their inherent instruction-following and reasoning abilities. Specifically, for SciCode,

Table 9: Performance comparison of EM-INF against test-time scaling methods. **Bold** & *Italics* indicates the best and second best performance, respectively. We report the mean $\pm$ standard deviation of four independent runs with different seeds. We use temperature, $t = 0.1$ and sample $N = 1$ trajectory unless specified.

| | Math | | | | | | Coding | Science |
|---|---|---|---|---|---|---|---|---|
| | **Math** | **AMC** | **AIME** | **Minerva** | **Olymp.** | **Avg.** | **LeetCode** | **UGPhysics** |
| Qwen2.5-Math-7B-Instruct | $80.7 \pm 1.2$ | $50.6 \pm 1.6$ | $11.9 \pm 0.7$ | $35.7 \pm 1.4$ | $\mathbf{41.1 \pm 0.8}$ | 44.0 | $0.6 \pm 0.3$ | $18.5 \pm 0.9$ |
| Greedy Decoding ($t = 0$) | $79.0 \pm 0.0$ | $49.4 \pm 0.0$ | $10.0 \pm 0.0$ | $34.6 \pm 0.0$ | $39.9 \pm 0.0$ | 42.6 | $1.1 \pm 0.0$ | $18.8 \pm 0.0$ |
| Iterative-refinement ($N = 3$) | $73.0 \pm 0.6$ | $39.7 \pm 0.9$ | $7.5 \pm 1.3$ | $32.4 \pm 1.8$ | $33.2 \pm 0.8$ | 37.2 | $0.4 \pm 0.2$ | $17.4 \pm 1.5$ |
| Self-consistency ($N = 4$) | $80.7 \pm 1.3$ | $50.9 \pm 1.1$ | $13.3 \pm 0.8$ | $\mathbf{36.7 \pm 0.8}$ | $41.0 \pm 0.3$ | *44.5* | $0.8 \pm 0.4$ | $18.5 \pm 0.3$ |
| ADAPTIVE TEMP | $\mathbf{81.9 \pm 1.4}$ | $53.1 \pm 0.7$ | $10.7 \pm 1.6$ | $36.3 \pm 1.6$ | $39.8 \pm 1.2$ | 44.4 | $\mathbf{2.2 \pm 0.2}$ | $\mathbf{22.5 \pm 1.0}$ |
| EM-INF | $81.3 \pm 0.7$ | $\mathbf{55.2 \pm 1.3}$ | $\mathbf{14.8 \pm 1.2}$ | $34.8 \pm 0.8$ | $40.6 \pm 0.6$ | $\mathbf{45.4}$ | $0.6 \pm 0.2$ | $21.4 \pm 1.2$ |
| Qwen2.5-7B-Instruct | $74.3 \pm 0.6$ | $44.6 \pm 1.1$ | $9.1 \pm 1.8$ | $35.2 \pm 0.9$ | $38.4 \pm 0.4$ | 40.3 | $47.2 \pm 1.3$ | $22.7 \pm 0.9$ |
| Greedy Decoding ($t = 0$) | $73.2 \pm 0.0$ | $42.2 \pm 0.0$ | $8.8 \pm 0.0$ | $33.8 \pm 0.0$ | $37.9 \pm 0.0$ | 39.2 | $46.1 \pm 0.0$ | $23.7 \pm 0.0$ |
| Iterative-refinement ($N = 3$) | $72.3 \pm 1.3$ | $40.9 \pm 0.5$ | $8.8 \pm 2.2$ | $34.8 \pm 0.4$ | $38.9 \pm 1.1$ | 39.1 | $50.0 \pm 0.5$ | $21.5 \pm 1.1$ |
| Self-consistency ($N = 4$) | $74.4 \pm 0.8$ | $43.7 \pm 0.3$ | $10.0 \pm 0.5$ | $35.1 \pm 1.2$ | $\mathbf{39.2 \pm 1.2}$ | 40.5 | $48.8 \pm 0.6$ | $23.6 \pm 0.1$ |
| ADAPTIVE TEMP | $\mathbf{74.9 \pm 1.3}$ | $\mathbf{49.8 \pm 1.1}$ | $10.0 \pm 1.0$ | $\mathbf{38.4 \pm 1.0}$ | $37.5 \pm 0.9$ | $\mathbf{42.2}$ | $49.4 \pm 0.9$ | $24.1 \pm 1.0$ |
| EM-INF | $74.0 \pm 0.9$ | $46.7 \pm 0.6$ | $\mathbf{11.7 \pm 1.7}$ | $36.5 \pm 1.2$ | $38.7 \pm 0.8$ | *41.5* | $\mathbf{51.7 \pm 0.7}$ | $\mathbf{25.4 \pm 1.2}$ |
| Llama-3.1-8B-Instruct | $41.9 \pm 1.1$ | $19.2 \pm 1.5$ | $3.3 \pm 1.6$ | $21.1 \pm 1.4$ | $16.5 \pm 0.6$ | 20.4 | $13.1 \pm 0.8$ | $17.1 \pm 0.7$ |
| Greedy Decoding ($t = 0$) | $40.6 \pm 0.0$ | $16.9 \pm 0.0$ | $3.3 \pm 0.0$ | $21.0 \pm 0.0$ | $16.0 \pm 0.0$ | 19.6 | $12.8 \pm 0.0$ | $16.1 \pm 0.0$ |
| Iterative-refinement ($N = 3$) | $41.3 \pm 0.3$ | $18.7 \pm 0.7$ | $2.5 \pm 1.1$ | $20.3 \pm 0.4$ | $16.0 \pm 0.6$ | 19.8 | $9.6 \pm 0.9$ | $18.3 \pm 0.8$ |
| Self-consistency ($N = 4$) | $43.3 \pm 0.4$ | $20.2 \pm 0.6$ | $2.8 \pm 1.1$ | $20.1 \pm 1.3$ | $\mathbf{18.1 \pm 0.7}$ | 20.9 | $12.8 \pm 1.7$ | $16.9 \pm 0.0$ |
| ADAPTIVE TEMP | $43.1 \pm 1.5$ | $22.7 \pm 1.1$ | $3.3 \pm 0.9$ | $20.0 \pm 1.6$ | $17.3 \pm 1.0$ | *21.3* | $\mathbf{16.3 \pm 1.5}$ | $\mathbf{19.9 \pm 1.8}$ |
| EM-INF | $\mathbf{43.9 \pm 1.2}$ | $\mathbf{23.5 \pm 1.7}$ | $3.3 \pm 1.4$ | $\mathbf{21.3 \pm 0.8}$ | $16.2 \pm 0.4$ | $\mathbf{21.6}$ | $14.1 \pm 0.3$ | $17.6 \pm 0.7$ |

we include a more powerful model, Qwen2.5-32B-Instruct, due to its superior capability in handling complex problems, allowing us to evaluate the performance of EM-INF on stronger models.

**Llama-3.1 Family**   Llama-3.1 [24] is the second model family included in this work to ensure the generalizability of our methods across different models. For both the entropy minimization post-training and inference-time experiments, we select Llama-3.1-8B-Instruct as the base model.

### D.4   FLOPs Computation

Following [63, 37], we estimate the training FLOPs as $6PD$ and inference FLOPs as $2PD$ where $P$ is the number of parameters in the model and $D$ is the number of tokens. We use early stopping based on the validation set accuracy for all methods, which significantly changes the FLOPs utilized.

- EM-FT: On each training step, we sample one trajectory from the model costing $2Pn$ where $n$ is the sequence length followed by training costing $6Pn$. The total cost per step is thus $8Pn$. We set sequence length to $4096$, batch size to $512$ and the training early stops at $40$ gradient steps costing $1.01 \times 10^{17}$ FLOPs for a 7B parameter model.

- EM-RL, GRPO, RLOO: None of the methods use a reward model. We use GRPO and RLOO with output verification only. We sample $N = 4$ trajectories costing $8Pn$, then we make a forward pass on the reference model costing another $8Pn$ followed by training costing $24Pn$. The total training cost per step is thus $40Pn$. We set sequence length to $4096$, batch size to $512$ and the training early stops at $100$ gradient steps costing $13 \times 10^{17}$ FLOPs for a 7B parameter model.

