# OpenReview forum: "The Unreasonable Effectiveness of Entropy Minimization in LLM Reasoning"
_NeurIPS.cc/2025/Conference — NeurIPS 2025 poster_

### Official Review · Reviewer_PKyh · 2025-07-02

**Clarity:** 3
**Significance:** 3
**Originality:** 3
**Rating:** 4
**Confidence:** 5

**Summary:**

This paper investigates entropy minimization as a standalone training and inference objective for improving large language models' performance on reasoning tasks without labeled data. The authors propose three approaches: EM-FT , EM-RL, and EM-INF. They demonstrate competitive performance with supervised methods on math, physics, and coding benchmarks.

**Questions:**

See weakness

**Ethical Concerns:**

["NO or VERY MINOR ethics concerns only"]

**Final Justification:**

This paper provides interesting insights into RL training. However, there remains questions on reproducibility and what's the underlying reason of the phenomenon.

**Limitations:**

yes

**Quality:**

3

**Strengths And Weaknesses:**

Strengths:
1. Authors present a very interesting observation that entropy minimization is an effective way to improve model performance on reasoning tasks.
2. Based on this discovery, authors propose three ways to improve the model, FT, RL, and inference time augmentation. Experimental results show that all these methods can significantly outperform the base model, and be comparable to supervised RL training.

Weakness:
1. While the empirical results are strong, the paper lacks deep theoretical analysis of why entropy minimization works so effectively. The core assumption that "confident models are more likely to be correct" is intuitive but not rigorously validated across different domains and model architectures.
2. The paper could benefit from discussing how entropy minimization can interact with supervised RL training, e.g. will combining two rewards give better results?
3. Some existing works such as [1] show that doing entropy maximization during RL can also improve the model's performance, which seems contradictory to this paper. It'd be helpful if authors can discuss this phenomenon.
4. The paper posits that EM is effective only when a base model possesses sufficient "latent reasoning capabilities," as evidenced by the performance gap between Qwen and Llama models. However, the argument is supported primarily by the end-task performance differences. The paper would be more impactful if it included a more direct analysis of what these capabilities are and how to quantify them.
5. Code is not provided, so it could be hard to reproduce the results.

[1] https://arxiv.org/abs/2504.20571

---

> ### Author Rebuttal · Authors · 2025-07-31
>
> **Base model capabilities**
> The reviewer raised a great point on the impact of the choice of the pretrained model on the performance improvements achieved through entropy minimization. We observe that the improvements from EM-FT and EM-RL on Llama-3.1-8B-Instruct are less substantial on math reasoning tasks compared to those on Qwen-2.5 (Sec. 5). The intuition behind using entropy minimization is based on the observation that models tend to be more accurate on examples for which they make predictions with higher confidence [4,5,6]. However, [7] reveal that entropy minimization based methods improves performance as long as the finetuning data align closely with the training data. As the fine-tuning data diverges from the training distribution, the model's performance diminishes. Building on this observation, we conjecture that entropy minimization is most beneficial when there is reasonable alignment between the pre-training and fine-tuning data.
>
> A common assumption in RL, and test time scaling for LLMs is that it is expected that the base models can already solve a problem in N trials, i.e., we require at least one successful trajectory to reinforce and compute the advantage [2]. In output-verified RL, this assumption is applied by using accuracy based filters during training, which filters out un-solvable problems based on the success rate over N trials. This hypothesis suggests that the base model must generate high-quality responses with reasonable probability and coverage over the downstream task. Extrapolating this assumption, entropy minimization via EM-RL, EM-FT and EM-INF sharpens the probability distribution over all responses based on the model's confidence. We conjecture that this sharpening mechanism is expected to be more useful on base models which show better scaling under best of N [8]. Table 1 (rebuttal) shows the percentage gain under best of n sampling (N=4) compared to greedy decoding (N=1) averaged over MATH, AMC, Minerva and OlympiadBench. We observe that Qwen-2.5-math-7b shows larger gains compared to LLaMA-3.1-8B-instruct when number of responses is increased from N=1 to N=4, indicating that Qwen is more likely to produce a correct solution. We hypothesize that better inference time scaling (e.g. best of n) is a likely cause for improvements from EM-RL [8].
>
> Table 1: Average performance on MATH, AMC, Minerva and OlympiadBench with greedy decoding (N=1) and best of N (N=4). We select the best response by using self-consistency, i.e., we pick the answer with the highest frequency
> |Model|Greedy (N=1)|Best of N (N=4)|% Gain
> |---|---|---|---|
> |Qwen-2.5-math-7b|23.2|26.4|13.4|
> |LLaMA-3.1-8B-instruct|19.5|21.1|7.5|
> |
>
> To further understand initial model choice, we finetune another base model- DeepSeek-Math-7B-Instruct [11] on the same data and training settings as Qwen-2.5-7B-Math (Sec. 3.3). Deepseek-7b-math-instruct has been finetuned on math-related tokens for 500B tokens on top of Deepseek-coder-7b by the model creators. We use the instruction tuned version which has been further finetuned using chat data. We fine tune deepseek-7b-math-instruct using entropy minimization (EM-RL) and evaluate it in Table 2 (rebuttal), we observe that entropy minimization improves performance over the Deepseek-math-7b-instruct by 2.5% on average on math tasks. Compared to LlaMA both, Qwen-2.5-math and DeepSeek-7b-math-instruct have undergone domain specific finetuning [11,12]. This observation suggests that gains from EM-RL are not only limited to Qwen, and can be seen on stronger models tuned for the downstream task.
>
> Table 2: Performance comparison of unsupervised EM-RL against RLOO for DeepSeek-Math-7B-Instruct. We use the setup defined in Sec. 3.3. Bold indicates performance improvement over RLOO.
> |Model|Math|AMC|Minerva|Olympiad|Avg.|
> |---|---|---|---|---|---|
> |Deepseek-math-7b-instruct|42|16.9|19.1|12.6|22.65|
> |RLOO|42.4|24.1|20.6|13.2|25.07|
> |EM-RL-token|**43.4**|18.1|**22.8**|13.6|24.47|
> |EM-RL-sequence|42.8|21.7|**21.7**|**14.4**|25.15|
>
> The success and limitations of our method highlight the importance of the capabilities of the pretrained models. We call for the inclusion of EM as a baseline in the evaluation of future post-pretraining and inference-time scaling algorithms, to better separate the contribution of algorithmic innovations from the intrinsic capabilities of the model itself (Line 58). We will update Sec.3.4 with these analyses and results to better contextualize model choice. We will also rephrase (Line 48, Sec. 6. Conclusion) to better acknowledge the impact of initial checkpoints on RL tuning.
>
> **Entropy minimization and Supervised RL**
> The reviewer raises an interesting parallel of combining entropy minimization with supervised RL. [3] shows that typical maximum entropy reinforcement learning techniques (e.g. PPO, GRPO, Soft actor-critic) which apply entropy maximization struggle on challenging control tasks where a precise subset of actions are required, and the remaining actions will lead to non-recoverable states and consequently low final rewards. They find that maximum entropy regularization can encourage the policy to select multiple similarly mediocre actions rather than the best one for short-term entropy benefits. Extrapolating this observation, reasoning tasks such as math and coding, require a precise sequence of actions to arrive at the correct solution. We conjecture that entropy minimization (EM-RL) along with supervised RL can encourage the policy to take the most precise action over the several mediocre choices.
>
> Furthermore, [1] find that supervised RL finetuning in its default setting (RLOO/GRPO) drops entropy during the early stages of training. Meanwhile, the policy’s validation performance presents an inverse trend, where it rises rapidly when training starts, and then saturates at a certain level. They find that entropy needs to be effectively maintained during training to avoid saturation of policy performance, as also observed in [11]. This observation suggests that, for a given performance level, a policy with higher entropy is preferred, since low entropy policies are difficult to further improve. Building on these observations, we conjecture that if the pre-trained model has converged and is at a higher entropy level, further entropy minimization through EM-RL can help as a post-training step, since there is more entropy to consume by EM-RL. We will add this insight to Sec. 3.4 of the paper.
>
> **Entropy maximization vs minimization:**
>
> Thank you for pointing out this interesting work! Similar to [9], we tried to maximize entropy, but observed that entropy maximization alone without the policy loss collapses the policy after 10 gradient steps while using Qwen-2.5-7B-Math, this observation was also noted in [9]. While [9] observes performance improvements, we conjecture this is because they train on a single carefully selected example.  Furthermore, [9] trains performed on test-set examples where making the model more random (larger entropy) could be desirable, similar to best of n [2]. We find that only negative entropy reward (entropy minimization) is sufficient for RL with similar effects without requiring any data selection. Standalone entropy maximization does not converge and eventually collapses, but EM-RL and EM-FT via entropy minimization converge and can be used to train on large datasets (60K prompts) without output verification. We will add this discussion to Sec.3 of the paper.
>
>
> **Code release**
> We will release the code for reproducibility. Since the rebuttal guidelines do not allow external links, we will attach it to our final version.
>
> [1] Cui, Ganqu, Yuchen Zhang, Jiacheng Chen, et al. The entropy mechanism of reinforcement learning for reasoning language models
>
> [2] Beirami, Ahmad, Alekh Agarwal, et al. "Theoretical guarantees on the best-of-n alignment policy
>
> [3] Zhang, Ruipeng, Ya-Chien Chang, and Sicun Gao. When Maximum Entropy Misleads Policy Optimization
>
> [4] Yves Grandvalet and Yoshua Bengio. Semi-supervised learning by entropy minimization.
>
> [5] Subhankar Roy, Aliaksandr Siarohin, et al. Unsupervised domain adaptation using feature-whitening and consensus loss.
>
> [6] Wang, D., Shelhamer, E., Liu, S., et al, T. Tent: Fully test-time adaptation by entropy minimization.
>
> [7] Press, Ori, Ravid Shwartz-Ziv, et al. The entropy enigma: Success and failure of entropy minimization
>
> [8] Huang, Audrey, Adam Block, et al  Self-improvement in language models: The sharpening mechanism
>
> [9] Wang, Yiping, Qing Yang, Zhiyuan Zeng, et al. Reinforcement learning for reasoning in large language models with one training example
>
> [10] Yang, An, Anfeng Li, Baosong Yang, Beichen Zhang, Binyuan Hui, Bo Zheng, Bowen Yu et al. Qwen3 technical report.
>
> [11] Shao, Zhihong, Peiyi Wang et al. Deepseekmath: Pushing the limits of mathematical reasoning in open language models
>
> [12] Yang, An, Beichen Zhang, et al. Qwen2. 5-math technical report: Toward mathematical expert model via self-improvement

---

> > ### Author Response · Authors · 2025-08-05
> >
> > We thank the reviewer for the thoughtful reviews and comments! We address reviewer comments above and incorporate all feedback. We are happy to have more discussions about any questions and concerns.

---

> > > ### Comment · Reviewer_PKyh · 2025-08-07
> > >
> > > Thank you for the responses. I will keep my scores.

---

### Official Review · Reviewer_3pzP · 2025-07-03

**Clarity:** 3
**Significance:** 4
**Originality:** 4
**Rating:** 6
**Confidence:** 3

**Summary:**

This paper presents a suite of related techniques for boosting the reasoning
performance of post-pre-training, pre-fine-tuning LLMs using entropy
minimization (EM).  In particular, it introduces two train-time methods,
EM-based "supervised" fine-tuning and EM-based RL fine-tuning as well as an
EM-based inference time algorithm.  The intuition is that after pre-training
LLMs already have strong priors and that decreasing entropy (uncertainty) will
lead stronger performance, particular in tasks that have definite answers and
require sequence of intermediate steps.  The results show that the EM-based
methods can indeed improve LLM performance reasoning tasks with lower compute
requirements and no need for labelled data.

**Questions:**

The paper is clear enough such that I do not have any major questions.

- How does this relate to the general principle the neural networks are
  over-confident and calibrating them usually involve strategically decreasing
  their confidence?
  - How does relate to hallucinations and over-confidence? I suppose reasoning
    tasks are maybe less hallucination-prone compared something like
    question-answering which it seems like EM-based methods are less likely to
    improve.
- Does this suggest that pre-training itself could benefit from incorporating
  some sort of entropy-maximization, or does EM only serve to benefit the model
  after pre-training has finished?
- For EM-INF, by "optimizing the logits" do you mean optimizing the layer that
  produces the logits?  By my understanding, the logits themselves are just
  values, and adjusting the values directly does not make sense (as opposed to
  adjusting, say weights/parameters).

### Minor points
- ? [Tab. 2] Does 60k unlabeled prompts mean the prompts from the particular task just without the provided answers?
- It seems like self-consistency could still be applied to long-output tasks if you use a fuzzy matching function instead of just an indicator (e.g., which output has the lowest average Levenshtein distance to all other outputs?).
- [Line 198] By "1e-6", do you mean "1 * 10^-6" as opposed to "1 * exp(-6)"?
- "chain of thoughts" -> "chains of thought"
- "lost in the sauce" is a bit too informal.

**Ethical Concerns:**

["NO or VERY MINOR ethics concerns only"]

**Final Justification:**

I maintain my positive assessment of the paper after the discussion and clarifications offered by the authors.

**Limitations:**

yes

**Paper Formatting Concerns:**

None.

**Quality:**

4

**Strengths And Weaknesses:**

### Strengths
- The results of the apply methods demonstrate frequent (maybe not consistent)
  improvements without the use of labelled data.  Insofar as this is shown to
  match performance to standard methods, it represents a Pareto-improvement
  since it eliminates the need for labeled data.
- A variety of related methods are presented which is not only useful insofar
  as each method has slightly different applicability, but also suggests
  something deeper is at play with entropy minimization.
- The presentation of the methods is clear and easy to follow.
### Weaknesses
- (minor) The experiments and analysis do not give a wholly clear picture of
  when EM-based methods will work versus when they won't.  These details could
  help indicate whether the proposed methods will be game-changers versus
  a niche way to squeeze out performance in particular scenarios.

---

> ### Author Rebuttal · Authors · 2025-07-31
>
> **When is entropy minimization helpful?**
>
> The reviewer raised a great point on when EM works. The intuition behind using EM is based on the observation that models tend to be more accurate on examples for which they make predictions with higher confidence (L19, L76). However, [1] reveals that entropy minimization based methods improves performance as long as the finetuning data align closely with the pretraining data. As this fine-tuning data diverges from the training distribution, the model's performance diminishes. Building on this observation, we conjecture that EM is most beneficial when there is reasonable alignment between the pre-training and fine-tuning data, as supported by model performance on downstream tasks.
>
> Furthermore, a common assumption in RL, and test time scaling for LLMs is that it is expected that the base models can already solve a problem in N trials, i.e., we require at least one successful trajectory to reinforce and compute the advantage [2]. In output-verified RL, this assumption is applied by using accuracy based filters during training, which filters out un-solvable problems based on the success rate over N trials. This hypothesis suggests that the base model must generate high-quality responses with reasonable probability and coverage over the downstream task. Extrapolating this assumption, EM-RL, EM-FT and EM-INF sharpens the probability distribution over all responses based on the model's confidence [4]. We conjecture that this sharpening mechanism is expected to be more useful on base models which show better scaling under best of N.
>
> We will update Sec.3.4 with this discussion to better contextualize when entropy minimization can help. We will also add this to our future work and limitations section and acknowledge that the success and limitations of entropy minimization is a great research direction.
>
>
> **Over-confidence and calibration**
>
> Thank you for raising this interesting point on over-confidence and calibration! Entropy minimization (EM-RL, EM-FT) encourages the model to produce low-entropy (i.e., highly confident) predictions. We conjecture that entropy minimization can both improve or hurt calibration. If entropy minimization helps the model become more confident in the right direction- that is, it becomes more confident on correctly generated solutions, then both performance and calibration improves. We expect that when the pre-trained model’s data distribution closely aligns with the fine-tuning task and the pre-trained model produces the correct response with a reasonable probability, entropy minimization should benefit performance and calibration. On the other hand, entropy minimization can harm calibration when it reinforces highly confident (low entropy) incorrect solutions to be generated with higher probability.
> We measure the expected calibration error (ECE) of both LLaMA and Qwen 2.5 models in Table 2 (rebuttal). We observe that the ECE for Qwen models improves, likely because EM-RL substantially benefits Qwen 2.5 math on math reasoning (Sec.3.4) . On the other hand, we notice that entropy minimization harms LLaMA’s calibration likely because we noticed smaller improvements on using EM-RL with LLaMA. We conjecture that the impact of entropy minimization on calibration is tied to the base model capability and expect that entropy minimization should likely benefit or have minimal impact on model performance and calibration.
>
> Table 2: Expected Calibration Error for EM-RL-sequence and EM-RL-token compared to the base model on the AMC math dataset. Lower ECE is better.
> |Model|ECE on AMC|
> |-|-|
> |Qwen2.5-Math-7B|64.9|
> |EM-RL-SEQUENCE $N=4$|43.5|
> |EM-RL-TOKEN $N=4$|44.7|
> |
> |Llama-3.1-8b-Instruct|79.0|
> |EM-RL-SEQUENCE $N=4$|81.5|
> |EM-RL-TOKEN $N=4$|83.3|
>
>
> **Non-reasoning tasks**
> We evaluate EM-INF and EM-RL on knowledge intensive tasks- UGPhysics and MMLU and observe that EM can improve performance UGPhysics, but maintains the same performance on MMLU.
>
> UGPhysics is a question answering benchmark for Physics which requires undergrad level physics knowledge [3]. We observe that entropy minimization with logit optimization (EM-INF) improves Qwen-2.5-7B-Instruct, Qwen-2.5-math-7B-Instruct, and Llama-3.1-8B-Instruct by 4% on average, indicating that unsupervised EM-INF helps improve pre-training performance by making the model more deterministic (Table 3). We will add this description about UGPhysics in Sec.4 to clarify.
>
> We additionally evaluate MMLU-stem-COT performance for our RL tuned models (EM-RL) in Table 3 (rebuttal). We observe that EM-RL did not impact its performance on knowledge intensive MMLU. This observation along with calibration errors, indicates that entropy minimization does not necessarily make the model too confident on non-reasoning tasks.
>
> Table 3: Performance of EM-RL compared with GRPO and RLOO on the MMLU-stem-COT dataset.
> |Model|MMLU-stem-CoT|
> |---|---|
> |Qwen2.5-Math-7B|63.6|
> |RLOO|63.2|
> |GRPO|64.1|
> |EM-RL-Sequence|63.7|
> |EM-RL-Token|64.4|
>
> **Entropy maximization during Pre-training**
> Entropy maximization during pre-training is an interesting idea to explore! [5] find that supervised RL finetuning in its default setting (e.g. GRPO) drops entropy during the early stages of training. Meanwhile, the policy’s validation performance presents an inverse trend, where it rises rapidly when training starts, and then saturates at a certain level. They find that entropy needs to be effectively maintained during training to avoid saturation of policy performance. This observation suggests that, for a given performance level, a policy with higher entropy is preferred, since low entropy policies are difficult to further improve. Building on these observations, we conjecture that if the pre-trained model has converged and is at a higher entropy level, further entropy minimization through EM-RL can help as a post-training step can benefit, since there is more entropy to consume by EM-RL. We will add this insight to Sec. 3.4 of the paper.
>
>
> **Logit optimization**
>
> EM-INF treats the model’s output logits as if they were free parameters, and uses gradient descent to update them to minimize the entropy of the distribution they induce; no gradient wrt. or the model parameters or parameter update is needed (Line 260). This optimization is performed during each decoding step to reduce the entropy of the LLM’s distribution without any parameter update. After logit optimization, we use sampling based decoding to select the next token. EM-INF can change the order of non-top logits (proposition 1), thereby generating different sequences. Empirically, we find that EM-INF works on complex tasks with high uncertainty (e.g. AIME math, UGPhysics and SciCode)(Table 3). By using EM-INF, we observe that Qwen 32B can outperform frontier models like GPT-4o on Scicode and is 3x more efficient than inference scaling through self-consistency and sequential refinement (Table 4, Figure 1).
>
> **[Tab. 2] Does 60k unlabeled prompts mean the prompts from the particular task just without the provided answers?**
>
> Yes, the 60k unlabeled prompts is a collection of prompts without the ground truth answers.
>
> **Self-consistency on long output tasks**
>
> Thank you for pointing out this interesting experiment! We agree that self-consistency can be applied using more sophisticated measures of consistency beyond exact matching, such as, fuzzy string match, embedding based similarity and using another LLM to measure consistency. We leave experimentation with these techniques as future work and we will clarify this throughout our paper to acknowledge this.
>
> We experiment with fuzzy matching for self-consistency for EM-INF in the Table 4. We find that our trend that entropy minimization (EM-INF, adaptive temperature) improves inference time scaling methods (self-consistency and iterative refinement) by 3% still holds. We will add these results to our paper and acknowledge that self-consistency can be performed on long outputs using methods like fuzzy string matching.
>
>
> Table 4: Performance comparison of EM-INF against test-time scaling methods on LeetCode. Bold & Italics indicates the best and second best performance, respectively. We use temperature, t= 0.1 and sample N = 1 trajectory unless specified. For self-consistency on long outputs, we pick the output which has the lowest average Levenshtein distance to all other outputs.
> | Model|LeetCode|
> |-|-|
> |Qwen2.5-Math-7B-Instruct|0.6|
> |Greedy Decoding ($t=0$)|1.1|
> |Iterative-refinement ($N=3$)|*1.7*|
> |Self-consistency ($N=4$)|0.8|
> |Adaptive Temp|**2.2**|
> |EM-INF|0.6|
> |
> |Qwen2.5-7B-Instruct|47.2|
> |Greedy Decoding ($t=0$)|46.1|
> |Iterative-refinement ($N=3$)|*50.0*|
> |Self-consistency ($N=4$)|48.3|
> |Adaptive Temp|49.4|
> |EM-INF|**51.7**|
> |
> |Llama-3.1-8B-Instruct|10.6|
> |Greedy Decoding ($t=0$)|12.8|
> |Iterative-refinement ($N=3$)|11.7|
> |Self-consistency ($N=4$)|14.9|
> |Adaptive Temp|**17.2**|
> |EM-INF|*14.4*|
>
>
> **"1e-6"**
> 1e-6 means 1*10^-6. We will replace L198 with this expression for clarity.
>
> **Typo and language**
> Thank you for pointing these out! We will fix the typo and replace "lost in the sauce” with: For example, problems requiring a complex, long chain of thought, where being more deterministic at every step helps prevent the model from losing track of the reasoning process.”
>
>
> [1] Press, Ori, Ravid Shwartz-Ziv, et al. The entropy enigma: Success and failure of entropy minimization
>
> [2] Beirami, Ahmad, Alekh Agarwal, et al. Theoretical guarantees on the best-of-n alignment policy
>
> [3] Xu, Xin, Qiyun Xu, Tong Xiao, et al Ugphysics: A comprehensive benchmark for undergraduate physics reasoning with large language models
>
> [4] Huang, Audrey, Adam Block, et al Self-improvement in language models: The sharpening mechanism
>
> [5] Cui, Ganqu, Yuchen Zhang et al. The entropy mechanism of reinforcement learning for reasoning language models.

---

> > ### Author Response · Authors · 2025-08-05
> >
> > We thank the reviewer for the thoughtful reviews and comments! We address reviewer comments above and incorporate all feedback. We are happy to have more discussions about any questions and concerns.

---

> > ### Comment · Reviewer_3pzP · 2025-08-07
> >
> > Thank you for the thorough reply.  My original question have been sufficiently addressed, and I maintain my positive assessment of the paper.

---

### Official Review · Reviewer_mT9o · 2025-07-03

**Clarity:** 4
**Significance:** 4
**Originality:** 3
**Rating:** 6
**Confidence:** 3

**Summary:**

This paper explores entropy minimization in the context of LLMs, particularly, as a way to improve performance without additional labeled data. The authors propose three methods to do so: minimizing token-level entropy directly, using entropy as a reward in reinforcement learning, and optimizing the logits to reduce entropy at inference time. Their experiments show that entropy minimization improves performance both on math and coding tasks.

**Questions:**

What decoding method do you when sampling responses from the base LM?

l162-4: this sentence is unclear to me. What does “lost in the sauce” mean?

Is the KL regularization term used for both EM-RL-token and EM-RL-sequence or just the former? If only the former, which it seems like from 3.2, why?

**Ethical Concerns:**

["NO or VERY MINOR ethics concerns only"]

**Final Justification:**

The (minor) weaknesses I pointed out in my review have been resolved and I maintain my recommendation to accept.

**Limitations:**

Appropriately addressed.

**Paper Formatting Concerns:**

None.

**Quality:**

3

**Strengths And Weaknesses:**

**Strengths:**
* I think this is an overall strong paper. It makes connections between an extensively studied ML method and large language models, which is interesting in itself, and additionally demonstrates promising results.
* The empirical evaluation is extensive; it is done on a diverse set of datasets and across several LLMs.
* Entropy minimization turns out to be useful both for math and coding tasks, which I think is an exciting finding given that it is an unsupervised method.
* The paper also discusses limitations and settings where the method does not work, namely, for tasks where confidence is not a good indicator of performance and for less capable base models.
* There is appropriate discussion of related work.


**Weaknesses:**
1. The method assumes that we take random samples from the LLM during decoding (I take it that entropy minimization won’t make much of a difference if you do greedy decoding), so naturally there is variance in the estimates. Yet, the authors only report point estimates in their experiment, i.e., there are no error bars. It would be great if that could be added to have more confidence in the results and ensure reproducibility.
2. Again, regarding variance: Trajectory-level entropy estimator must have very high variance, and N=4 rollouts therefore seems very small. I’m surprised that it works with such a small number of rollouts, but it would nevertheless be interesting to see an analysis on how this impacts performance, and to understand if there is a trade-off between performance and runtime.
3. Tab 3 shows that adaptive temperature, which is a simpler method, often outperforms the proposed inference method based on entropy. I think that should be conveyed more clearly in the text in order not to mislead.
4. It's a bit odd that Prop. 1 is only given in an informal manner. I don't think it would be too complicated to write out formally what it is you prove.

---

> ### Author Rebuttal · Authors · 2025-07-31
>
> **Error bars**
>
> We thank the reviewer for the great suggestion. We provide all result tables with error bars below by running the evaluation three times with a different random seed. We report the mean $\pm$ standard deviation for all models. We observe that our trend is still consistent with EM-RL, without any labeled data, achieving comparable or better performance than RL baselines such as GRPO and RLOO that are trained on 60K labeled examples. We will add these error bars to aid clarity and reproducibility in the paper. Furthermore, our trends with inference time logit optimization (EM-INF) is still consistent with improving the performance of the base model for almost all tasks and model classes by 3% on average.
>
> Table 1: Performance comparison of unsupervised finetuning (EM-FT) and various rewarding methods in EM-RL with supervised finetuning and RL. Italics, Bold indicates performance improvement over GRPO and SC-RL (self-consistency RL), respectively.
>
> |Model|Math|AMC|AIME|Minerva|Olymp.|Avg.|LeetC|LiveC|Avg.|FLOPs|
> |-|-|-|-|-|-|-|-|-|-|-|
> |**Qwen2.5-7b**|43.8($\pm1.1$)|31.3($\pm2.7$)|15.6($\pm3.7$)|14.7($\pm1.0$)|19.0($\pm1.5$)|24.9|26.1($\pm1.8$)|18.4($\pm0.9$)|22.3|–|
> |
> |w/SFT N=1|48.2($\pm2.4$)|30.2($\pm2.6$)|10.0($\pm0.6$)|17.6($\pm2.3$)|22.4($\pm1.7$)|25.7|18.3|18.3|18.3|1.0|
> |w/RLOO N=4|73.0($\pm0.9$)|57.8($\pm2.9$)|23.3($\pm4.8$)|31.2($\pm0.6$)|34.2($\pm0.7$)|43.9|28.3|26.7|27.5|13.1|
> |w/GRPO N=4|71.8($\pm0.6$)|56.6($\pm2.0$)|21.1($\pm2.4$)|25.0($\pm3.6$)|35.9($\pm1.0$)|42.1|25.0|25.8|25.4|13.1|
> |
> |EM-FT N=1|67.2($\pm0.6$)|51.8($\pm1.0$)|14.4($\pm2.0$)|*33.1($\pm0.7$)*|*34.4($\pm0.2$)*|40.2|*28.3($\pm1.1$)*|17.2($\pm0.4$)|22.8|1.0|
> |SC-RL N=4|73.2($\pm0.6$)|51.8($\pm2.2$)|15.6($\pm1.6$)|*26.1($\pm1.0$)*|*36.7($\pm0.5$)*|40.7|–|–|–|13.1|
> |EM-RL-SEQUENCE N=4|67.2($\pm1.4$)|**53.0($\pm2.1$)**|**21.1($\pm2.8$)**|***30.9($\pm2.3$)***|35.6($\pm0.9$)|**41.6**|*31.1($\pm1.4$)*|21.7($\pm1.1$)|*26.4*|13.1|
> |EM-RL-TOKEN N=4|70.8($\pm0.8$)|***57.8($\pm2.6$)***|**18.9($\pm3.1$)**|***30.9($\pm1.0$)***|35.9($\pm0.6$)|**42.9**|*29.5($\pm0.9$)*|24.5($\pm0.7$)|*27.0*|13.1|
>
> Table 2:Performance comparison of EM-INF against test-time scaling methods. Bold & Italics indicates the best and second best performance, respectively. Dash line ("–") denotes that selfconsistency is inapplicable. We use temperature, t= 0.1 and sample N = 1 trajectory unless specified
> |Model|Math|AMC|AIME|Minerva|Olymp.|Avg.|LeetCode|UGPhysics|
> |-|-|-|-|-|-|-|-|-|
> |Qwen2.5-Math-7B-Instruct|79.2($\pm0.7$)|48.2($\pm1.6$)|8.9($\pm0.6$)|**43.4($\pm1.2$)**|**41.3($\pm0.6$)**|44.2|0.6($\pm0.4$)|18.6($\pm1.0$)|
> |Greedy Decoding($t=0$)|79.0|53.0|*11.1*|*42.6*|39.9|45.1|1.1|18.8|
> |Iterative-refinement($N=3$)|79.2($\pm0.0$)|48.2($\pm0.0$)|8.9($\pm0.0$)|**43.4($\pm1.1$)**|**41.3($\pm0.6$)**|44.2|*1.7($\pm0.2$)*|16.4($\pm0.8$)|
> |Self-consistency($N=4$)|78.8($\pm0.0$)|53.0($\pm0.0$)|*11.1($\pm0.0$)*|**43.4($\pm1.2$)**|40.7($\pm0.7$)|*45.4*|0.8($\pm0.4$)|19.6($\pm0.8$)|
> |Adaptive Temp|**80.8($\pm0.9$)**|**55.4($\pm0.9$)**|10.0($\pm1.9$)|41.5($\pm2.1$)|38.7($\pm1.2$)|45.3|**2.2($\pm0.3$)**|**23.7($\pm0.7$)**|
> |EM-INF|*80.2($\pm0.6$)*|*54.2($\pm2.5$)*|**14.4($\pm1.6$)**|42.3($\pm0.8$)|*40.8($\pm0.6$)*|**46.4**|0.6($\pm0.4$)|*23.1($\pm1.3$)*|
> |
> |Qwen2.5-7B-Instruct|74.0($\pm0.9$)|41.0($\pm1.8$)|8.9($\pm0.9$)|*41.2($\pm0.7$)*|37.5($\pm0.2$)|40.5|47.2($\pm1.5$)|23.4($\pm1.3$)|
> |Greedy Decoding($t=0$)|*74.2*|44.6|7.8|40.4|37.2|40.8|46.1|23.7|
> |Iterative-refinement($N=3$)|73.8($\pm0.0$)|37.3($\pm0.0$)|*10.0($\pm0.0$)*|41.5($\pm0.7$)|37.8($\pm0.2$)|40.1|*50.0($\pm0.5$)*|21.5($\pm1.2$)|
> |Self-consistency($N=4$)|73.4($\pm0.0$)|43.4($\pm0.0$)|*10.0($\pm0.0$)*|33.5($\pm2.9$)|**41.3($\pm0.8$)**|40.3|48.3($\pm0.9$)|23.4($\pm1.4$)|
> |Adaptive Temp|**74.4($\pm0.8$)**|**50.6($\pm1.8$)**|*10.0($\pm1.6$)*|**41.9($\pm0.9$)**|*39.7($\pm0.7$)*|**43.3**|49.4($\pm0.9$)|*25.8($\pm1.0$)*|
> |EM-INF|*73.8($\pm1.1$)*|*45.8($\pm0.9$)*|**11.1($\pm0.5$)**|*41.2($\pm0.7$)*|38.2($\pm0.5$)|*42.0*|**51.7($\pm0.7$)**|**26.9($\pm1.1$)**|
> |
> |Llama-3.1-8B-Instruct|40.6($\pm0.5$)|18.1($\pm1.5$)|1.1($\pm1.6$)|22.4($\pm1.0$)|15.7($\pm0.8$)|19.6|10.6($\pm1.2$)|17.5($\pm0.7$)|
> |Greedy Decoding($t=0$)|40.6|16.9|3.3|21.0|16.0|19.6|12.8|16.1|
> |Iterative-refinement($N=3$)|41.0($\pm0.0$)|19.3($\pm0.0$)|1.1($\pm0.0$)|22.4($\pm1.5$)|15.7($\pm0.8$)|19.9|11.7($\pm1.1$)|18.3($\pm0.6$)|
> |Self-consistency($N=4$)|41.2($\pm0.0$)|20.5($\pm0.0$)|*4.4($\pm0.0$)*|*20.2($\pm0.8$)*|**19.4($\pm0.4$)**|21.1|14.9($\pm1.1$)|*20.2($\pm1.1$)*|
> |Adaptive Temp|**43.6($\pm0.6$)**|**25.3($\pm1.8$)**|**5.5($\pm1.2$)**|**24.3($\pm1.1$)**|*16.6($\pm0.6$)*|**23.1**|**17.2($\pm1.7$)**|19.9($\pm1.9$)|
> |EM-INF|*43.0($\pm1.2$)*|*22.9($\pm2.3$)*|3.3($\pm1.4$)|*22.8($\pm0.7$)*|16.4($\pm0.4$)|*21.7*|*14.4($\pm0.3$)*|**20.4($\pm1.3$)**|
>
>
> **Sensitivity to number of rollouts $N$**
>
> The reviewer has a great observation that the trajectory level entropy estimator has a high variance. We set $N=4$ building on existing RL fine tuning works to make sure that the training overhead is consistent [1]. We measure the impact of the number of rollouts, N on EM-RL-sequence in Table 4 (rebuttal). We observe that EM-RL-sequence benefits with larger $N$ on MATH-500, suggesting that larger number of rollouts might reduce the variance. However, we note that a larger N does not help AMC, Minerva and Olympiad bench, and reduces performance on AIME. Overall, we observe consistent performance of EM-RL-sequence on sweeping N from 4 to 16. We will add these results to Sec. 3.4 of the paper as a plot.
>
> Table 4: Sensitivity to number of rollouts N of EM-RL-sequence. We use Qwen-2.5-Math-base and use the training setup defined in Sec. 3.3.
> |Model|Math|AMC|AIME|Minerva|OlympiadBench|Avg|
> |---|---|---|---|---|---|---|
> |EM-RL-sequence-N=4|67.2|53.0|21.1|30.9|35.6|41.56|
> |EM-RL-sequence-N=8|71.4|54.2|13.3|31.6|35.0|41.1|
> |EM-RL-sequence-N=16|70.6|51.9|15.5|23.9|35.1|39.4|
>
>
> **Effectiveness of adaptive temperature**
>
> Thank you for this suggestion. We will update Sec.4 results by adding the following:  Adapting the temperature (adaptive temperature) at every decoding step to reduce the entropy boosts model performance by 3% on average. Adaptive temperature is a simple method which is broadly applicable to all tasks while being computationally efficient.
>
> **Formal proof for prop 1**
>
> Thank you for pointing this out! A more formal proof is provided below. We will add this to Sec. 4.
>
> Proposition: Both adaptive temperature scaling and logit optimization reduce the entropy of a model’s output distribution. However, in high-uncertainty settings (i.e., when the model’s predictive distribution has high entropy), logit optimization can change the order of non-top logits. In contrast, temperature scaling preserves the order of logits and merely sharpens or flattens the distribution proportionally. Both temperature scaling and logit optimization will keep the token with the top logit and increase its probability, making them less likely to change the model’s behavior when it is highly confident.
>
> Proof:
> **Part 1:** $\forall \textbf{z} \in \mathbb{R}^{|\mathcal{V}|}$ (logits) let $p(\textbf{z}) = \text{softmax}(\textbf{z})$. Let $\mathcal{H}(p)$ denote the entropy of $p$;  $\mathcal{H}=-\sum_{i=1}^{|\mathcal{V}|}p_i \log p_i$.
> $\exists \textbf{z}$, $\eta>0$ and $a,b\in[1,\dots,|V|]$, such that
> $\textbf{z}_a<\textbf{z}_b$ and $\textbf{y}_a>\textbf{y}_b$ where $\textbf{y}_i = z_i + \eta p_i(\log p_i + \mathcal{H})$, $i=1,\dots,|\mathcal{V}|$.
>
>
> Consider $\mathbf{y_a}-\mathbf{y_b} = \mathbf{z_a}-\mathbf{z_b} + \eta (g_a-g_b)$ where $g_i = p_i (\log p_i + \mathcal{H})$. $g$ is the gradient of entropy $\mathcal{H}$ w.r.t. $\mathbf{z}$.
>
> We now show that $\exists \mathbf{z}, a,b$ such that $z_a<z_b$ and $g_a>g_b$.
>
> By Mean-Value Theorem $\exists c \in (p_a,p_b)$ such that $g_a-g_b = (p_a-p_b)(\log c + \mathcal{H} + 1)$.
> When $p_a-p_b < e^{-(\mathcal{H}+1)}$ we have
>
> $c<p_b<e^{-(\mathcal{H} + 1)} \implies \log c + \mathcal{H} + 1 < 0 \implies g_a-g_b > 0$
>
> Thus, we can set an arbitrary large $\eta$ such that,
> $z_a-z_b + \eta(g_a-g_b) > 0$
>
>
>
> **Part 2**: $b$ cannot be $\arg \max_{i} z_i$.
>
> Proof: Let $j = \arg\max_i \mathbf{z_i}$, i.e. the index of the maximum logit.
> Since $p_j = \max_i p_i$ we can bound it using the inequality:
>
> $\mathcal{H} (p) = -\sum_i p_i \log p_i \geq -\log p_j \quad \Rightarrow \quad p_j \geq e^{-\mathcal{H}(p)}\geq e^{-(\mathcal{H} (p)+1)}$
>
> We now analyze how $g_i$ behaves as a function of $p_i$. Let $g(p_i) = p_i (\log p_i + \mathcal{H})$ and $g'(p_i) = \log p_i + \mathcal{H} + 1$.
> Thus, $g(p_i)$ is monotonically increasing whenever $g'(p_i)>0$, i.e. when,
> $p_i > e^{-(\mathcal{H} + 1)}$
> So no lower-probability index $i$ can have $g_i>g_j$, i.e., the entropy gradient cannot reduce $\mathbf{y_j}$ relative to others. Therefore, $b\neq j$, leading to the following behavior
>
> $0<g_a\leq g_b  \quad p_a \in (e^{-\mathcal{H}},p_b]$
>
> $g_a < 0 < g_b \quad p_a \in (0,e^{-\mathcal{H}})$
>
>
> **Decoding method for sampling?**
>
> We use sampling based decoding with a temperature of 0.1 for Table 2. We will add this to the Table caption to aid reproducibility.
>
> **l162-4: this sentence is unclear to me. What does “lost in the sauce” mean?**
>
> Thank you for pointing this out. We will rephrase it as the following to make it more formal- “​​For example, problems requiring a complex, long chain of thought, where being more deterministic at every step helps prevent the model from losing track of the reasoning process.”
>
> **Is the KL regularization term used for both EM-RL-token and EM-RL-sequence or just the former?**
>
> The KL term is used in both EM-RL-token and EM-RL-sequence. We will add this to Sec. 3.2 more explicitly.
>
> [1] Cui, Ganqu, Lifan Yuan,  et al. Process reinforcement through implicit rewards

---

> > ### Comment · Reviewer_mT9o · 2025-08-01
> >
> > Thank you for the clarifications and for making the suggested improvements; this is great.

---

### Official Review · Reviewer_AvWR · 2025-07-03

**Clarity:** 3
**Significance:** 4
**Originality:** 4
**Rating:** 5
**Confidence:** 3

**Summary:**

The paper demonstrates that Entropy Minimization can effectively improve the performance of large language models on complex math, physics, and coding tasks—even without using any labeled data.

**Questions:**

1. What is j in Formula 5 (Lines 265 -266)

**Ethical Concerns:**

["NO or VERY MINOR ethics concerns only"]

**Final Justification:**

Most of my concerns during rebuttal have been addressed, so I have updated my score accordingly.

**Limitations:**

Yes

**Quality:**

3

**Strengths And Weaknesses:**

**Strengths**:

1. The paper is well-structured, and the core idea is both compelling and original, offering a promising new direction for unlocking the potential of language models in post-training.

2. The greatest strength of Entropy Minimization lies in its independence from large amounts of high-quality labeled data, eliminating the need to collect and curate such annotations. This makes it a more flexible and efficient approach to post-training, with significant potential for broader application.


**Weaknesses**:

1. Why do these improvements appear only on the Qwen model? Could the authors provide more insights by analyzing the internal parameter characteristics or pretraining data of Qwen? Why does the same approach not work as well on LLaMA?

2. Beyond math and coding tasks, have the authors evaluated performance on other domains, such as memory-intensive tasks?

3. Regarding the KL regularization coefficient β, how does it affect training outcomes? It would be helpful to see additional training results and analysis related to its impact.

4. I would like to see more CoT examples generated after Entropy Minimization training. How do these differ in style and structure from those produced by RL training with GRPO? And will EM fail to activate the model's self-verification capabilities as effectively as GRPO does?

If these concerns are properly addressed, I would be willing to reconsider my evaluation.

---

> ### Author Rebuttal · Authors · 2025-07-31
>
> **Impact of initial model choice**
>
> The reviewer raised a great point on the impact of the choice of the pretrained model on the performance improvements achieved through entropy minimization. We observe that the improvements from EM-FT and EM-RL on Llama-3.1-8B-Instruct are less substantial on math reasoning tasks compared to those on Qwen-2.5 (Sec. 5). The intuition behind using EM is based on the observation that models tend to be more accurate on examples for which they make predictions with higher confidence [6,7,8] (L19, L76). However, [9] reveals that EM based methods improves performance as long as the finetuning data align closely with the pretraining data. As this fine-tuning data diverges from the training distribution, the model's performance diminishes. Building on this observation, we conjecture that EM is most beneficial when there is reasonable alignment between the pre-training and fine-tuning data.
>
> A common assumption in RL, and test time scaling for LLMs is that it is expected that the base models can already solve a problem in N trials, i.e., we require at least one successful trajectory to reinforce and compute the advantage [10]. In output-verified RL, this assumption is applied by using accuracy based filters during training, which filters out un-solvable problems based on the success rate over N trials [11,12]. This hypothesis suggests that the base model must generate high-quality responses with reasonable probability and coverage over the downstream task. Extrapolating this assumption, entropy minimization via EM-RL, EM-FT and EM-INF sharpens the probability distribution over all responses based on the model's confidence [13]. We conjecture that this sharpening mechanism is expected to be more useful on base models which show better scaling under best of N. Table 1 (rebuttal) shows the percentage gain under best of n sampling (N=4) compared to greedy decoding (N=1) averaged over MATH, AMC, Minerva and OlympiadBench. We observe that Qwen-2.5-math-7b shows larger gains compared to LLaMA-3.1-8B-instruct when the number of responses is increased from N=1 to N=4, indicating that Qwen-2.5 is more likely to produce a correct solution. We hypothesize that better inference time scaling (e.g. best of n) is a likely cause for improvements from EM-RL, since we observe larger improvements on Qwen2.5 compared to LLaMA on math (Sec.5).
>
> Table 1: Average performance on MATH, AMC, Minerva and OlympiadBench with greedy decoding (N=1) and best of N (N=4). We select the best response by using self-consistency, i.e., we pick the answer with the highest frequency.
>
> |Model|Greedy (N=1)|Best of N (N=4)|% Gain
> |---|---|---|---|
> |Qwen-2.5-math-7b|23.2|26.4|13.4|
> |LLaMA-3.1-8B-instruct|19.5|21.1|7.5|
>
> To further understand initial model choice, we finetune another base model- DeepSeek-Math-7B-Instruct on the same data and training settings as Qwen-2.5-7B-Math (Sec.3.3). Deepseek-7b-math-instruct has been finetuned on math-related tokens for 500B tokens on top of Deepseek-coder-7b by the model creators [1]. We use the instruction tuned version which has been further finetuned using chat data. We fine tune deepseek-7b-math-instruct using entropy minimization (EM-RL) and evaluate it in Table 2 (rebuttal), we observe that EM improves performance over the Deepseek model by 2.5% on average on math tasks. Compared to LlaMA both, Qwen-2.5-math and DeepSeek-7b-math have undergone domain specific math finetuning [2,1]. This observation suggests that gains from EM-RL are not only limited to Qwen, and can be seen on stronger models finetuned for the downstream task.
>
> Table 1: Performance comparison of unsupervised EM-RL against supervised RLOO. Bold indicates performance improvement over RLOO. We use the same training setup defined in Sec. 3.3.
>
> |Model|Math|AMC|Minerva|Olympiad|Avg.|
> |---|---|---|---|---|---|
> |Deepseek-math-7b-instruct|42|16.9|19.1|12.6|22.65|
> |RLOO|42.4|24.1|20.6|13.2|25.07|
> |EM-RL-token|43.4|18.1|22.8|13.6|24.47|
> |EM-RL-sequence|42.8|21.7|21.7|14.4|25.15|
>
> The success and limitations of our method highlight the importance of the capabilities of the pretrained models. We call for the inclusion of EM as a baseline in the evaluation of future post-pretraining and inference-time scaling algorithms, to better separate the contribution of algorithmic innovations from the intrinsic capabilities of the model itself (Line 58). We are unable to analyze the pre-training data of either LLaMA and Qwen models, since they have not been released and acknowledge that studying the impact of base model choice in RL is of great importance. We will update Sec.3.4 with these analyses and results to better contextualize model choice. We will also rephrase (Line 48, Sec. 6. Conclusion) to better acknowledge the impact of initial checkpoints on RL tuning.
>
>
> **Evaluation on knowledge intensive tasks**
> We evaluate EM-INF and EM-RL on knowledge intensive tasks- UGPhysics and MMLU and observe that EM can improve performance UGPhysics, but maintains the same performance on MMLU.
>
> UGPhysics is a question answering benchmark for Physics which requires undergrad level physics knowledge [3]. We observe that entropy minimization with logit optimization (EM-INF) improves Qwen-2.5-7B-Instruct, Qwen-2.5-math-7B-Instruct, and Llama-3.1-8B-Instruct by 4% on average, indicating that unsupervised EM-INF helps improve pre-training performance by making the model more deterministic (Table 3). We will add this description about UGPhysics in Sec.4 to clarify.
>
> We additionally evaluate MMLU-stem-COT [4] performance for our RL tuned models (EM-RL) in Table 3 (rebuttal). We observe that EM-RL did not impact its performance on knowledge intensive MMLU.
>
> Table 3: Performance of EM-RL compared with GRPO and RLOO on the MMLU-stem-COT dataset [4].
> |Model|MMLU-stem-CoT|
> |---|---|
> |Qwen2.5-Math-7B|63.6|
> |RLOO|63.2|
> |GRPO|64.1|
> |EM-RL-Sequence|63.7|
> |EM-RL-Token|64.4|
>
>
> **Sensitivity to KL regularization coefficient $\beta$**
>
> We provide results with different $beta$ for EM-RL token below in Table 4 (rebuttal). We observe that a small $\beta$ value is better for entropy minimization because it weighs EM more in the overall loss function (L172).
>
> Theoretically, the KL regularization can increase the entropy when the base model is too uncertain, i.e., it deviates from the pre-trained checkpoint causing it to visit high entropy states. However when $\beta<1$ (we use $0.001$) (L172) the dominant gradient in the policy gradient is from entropy minimization. Since we set $\beta$ to a small value the overall objective minimizes the entropy.
>
> Table 4: Sensitivity of EM-RL-token to various values of the KL regularization coefficient.
> |Model|Math|AMC|AIME|Minerva|OlympiadBench|Avg.|
> |---|---|---|---|---|---|---|
> |Qwen-2.5-math-7b|43.8|31.3|15.5|14.7|19|24.86|
> |EM-RL-token-$\beta=1.5$|45|39.8|13.3|9.6|21.2|25.78|
> |EM-RL-token-$\beta=0.5$|50.8|39.8|16.7|10.7|23|28.2|
> |EM-RL-token-$\beta=0.001$|70.8|57.8|18.9|30.9|19|39.48|
> |EM-RL-token-$\beta=0$|70.9|57.5|18.2|29.6|19|39.04|
>
>
> **What is j in Formula 5**
> Thank you for pointing this out! $j$ indexes over the vocabulary, i.e., $j$ ranges from 1 to the vocabulary size. We will correct the notation to $L_{EM-INF} = max (-\sum_{j=1}^{|{V}|}\sigma(z_t)_j\log \sigma(z_t)_j,\delta)$
>
> **COT example**
>
> We provide example generations with entropy minimization at inference time (EM-INF) via logit optimizations in the Appendix (Figure 4). For EM-INF, we observe that the model produces more deterministic and concise outputs, since entropy minimization optimizes the model to seek a specific mode. This behaviour helps improve on complex/highly uncertain tasks such as scientific coding.
>
> We provide EM-RL generations below and compare it with regular RLOO model generations. We observe that both EM-RL and RLOO invoke the self-verification step in their generations. These observations indicate that the entropy minimization makes the model more deterministic and generates a more concise outputs while preserving self-verification. We will add these qualitative analysis to the results section.
>
> **RLOO**
> [omit] **Let's check if this solution works.** The five integers are 1, 8, 10, 10. [omit] The median is 2 greater than the arithmetic mean, [omit] So, the solution works. The least possible value for the mode is 10.
>
> **EM-RL-seq**
> [omit] **We should verify that this solution works by checking the values of a, b, c, d and e**
> The median is [omit] 2 greater than the median. [omit] So, the solution is correct. Thus, the least possible value for the mode is 10.
>
> [1] Shao, Zhihong, Peiyi Wang et al. Deepseekmath: Pushing the limits of mathematical reasoning in open language models
>
> [2] Yang, An, Beichen Zhang, et al. Qwen2. 5-math technical report: Toward mathematical expert model via self-improvement
>
> [3] Xu, Xin, Qiyun Xu, et al Ugphysics: A comprehensive benchmark for undergraduate physics reasoning with large language models
>
> [4] Hendrycks, Dan, Collin Burns, et al. Measuring massive multitask language understanding
>
> [5] MMLU-Stem-COT subset from Deepseek evaluator: https://github.com/deepseek-ai/DeepSeek-Math
>
> [6] Yves Grandvalet and Yoshua Bengio. Semi-supervised learning by entropy minimization
>
> [7] Subhankar Roy, Aliaksandr Siarohin, et al. Unsupervised domain adaptation using feature-whitening and consensus loss
>
> [8] Wang, D., Shelhamer, E., Liu, S., et al, T. Tent: Fully test-time adaptation by entropy minimization
>
> [9] Press, Ori, Ravid Shwartz-Ziv, et al. The entropy enigma: Success and failure of entropy minimization
>
> [10] Beirami, Ahmad, Alekh Agarwal, et al. Theoretical guarantees on the best-of-n alignment policy
>
> [11] Yu, Qiying, Zheng Zhang, et al. Dapo: An open-source llm reinforcement learning system at scale
>
> [12] Cui, Ganqu, Lifan Yuan,  et al. Process reinforcement through implicit rewards
>
> [13] Huang, Audrey, Adam Block, et al Self-improvement in language models: The sharpening mechanism

---

> > ### Author Response · Authors · 2025-08-05
> >
> > We thank the reviewer for the thoughtful reviews and comments! We address reviewer comments above and incorporate all feedback. We are happy to have more discussions about any questions or concerns.

---

> > ### Comment · Reviewer_AvWR · 2025-08-07
> >
> > Thank you for the rebuttal and the additional experiments provided.
> >
> > Most of my concerns have been addressed, so I have updated my score accordingly.

---

### Note · Authors · 2025-08-16

We thank the reviewers for their thoughtful comments. We are encouraged to know that they find our work well-written (AvWR, 3pzP), our unsupervised method effective (PKyh), original and compelling (AvWR), our connections to prior ML and LLM literature interesting (mT9o, 3pzP) and that our experiments exciting ( mT9o), effective (PKyh) which can enable the method to be a significant potential for broader application (AvWR). We incorporate all reviewer feedback in our paper.

One of the concerns was regarding when and why (3pzP, PKyh) entropy minimization was beneficial, and its relation with the base model (AvWR, PKyh). We tie entropy minimization (EM) for LLMs with traditional semi-supervised learning and test time adaptation to explain its success and limitations. Our method is based on the observation that models tend to be more accurate on examples for which they make predictions with higher confidence. We also connect performance gains from supervised and unsupervised RL with base model capabilities, by showing that better base model inference time scaling might result in larger improvements from EM analogous to the observation that correlated confidence and accuracy benefits entropy minimization. We provide results on an additional base model (DeepSeek 7B) to support that the effectiveness of EM may not be limited to only Qwen and evaluate the models on additional knowledge intensive tasks. We also test the models sensitivity to KL weighing coefficient, number of trajectories sampled, variance in results, and entropy minimizations effect on calibration and provide case studies showing similar CoTs when using supervised and unsupervised RL. Lastly, we discuss connections of EM to supervised RL, entropy maximization during RL tuning and pre-training, over-confidence and calibration. We are glad to find that these concerns were addressed sufficiently (AvWR, mT9o, 3pzP).

---

### Decision · Program_Chairs · 2025-09-17

**Decision:**

Accept (poster)

**Comment:**

This paper provides an in-depth investigation of Entropy Minimization (EM) for enhancing large language model performance, with its core advantage being independence from additional labeled data. The authors construct a comprehensive methodological framework encompassing:  EM-based "supervised" fine-tuning,  EM-based RL fine-tuning, and EM-based inference time algorithm, accompanied by exhaustive comparative analysis. Experimental results convincingly demonstrate EM's significant effectiveness in mathematical and coding tasks, offering a novel approach to strengthening LLM reasoning capabilities. The proposal to incorporate EM as a new baseline in future LLM reasoning evaluations represents a valuable contribution. With rigorous arguments, well-supported evidence, and effective responses to reviewers' concerns during discussion that demonstrated strong persuasiveness, the paper has earned unanimous approval. Therefore, I recommend accepting this paper.